# A Review of Electrical Assisted Photocatalytic Technologies for the Treatment of Multi-Phase Pollutants

**Chung-Shin Yuan** [1,*], **Iau-Ren Ie** [1], **Ji-Ren Zheng** [1], **Chung-Hsuan Hung** [2], **Zu-Bei Lin** [1] **and Ching-Hsun Shih** [1]

[1] Institute of Environmental Engineering, National Sun Yat-Sen University, Kaohsiung 80424, Taiwan; peter-34@yahoo.com.tw (I.-R.I.); april133346@gmail.com (J.-R.Z.); becky11663333@g-mail.nsysu.edu.tw (Z.-B.L.); x3413415@gmail.com (C.-H.S.)

[2] Department of Safety, Health and Environmental Engineering, National Kaohsiung University of Science and Technology, Kaohsiung 82445, Taiwan; jeremyh@nkust.edu.tw

* Correspondence: ycsngi@mail.nsysu.edu.tw; Tel.: +886-7-5252000 (ext. 4409); Fax: +886-7-52524409

**Abstract:** This article reviews the fundamental theories and reaction mechanisms of photocatalytic technologies with the assistance of electrical field for degrading multi-phase pollutants. Photo(electro)catalysis including photocatalytic oxidation (PCO) and photoelectrocatalytic oxidation (PECO) have been a potential technologies applied for the treatment of organic and inorganic compounds in the wastewaters and waste gases, which has been treated as a promising technique by using semiconductors as photo(electro)catalysts to convert light or electrical energy to chemical energy. Combining photocatalytic processes with electrical field is an option to effectively decompose organic and inorganic pollutants. Although photocatalytic oxidation techniques have been used to decompose multi-phase pollutants, developing efficient advanced oxidation technologies (AOTs) by combining photocatalysis with electrical potential is urgently demanded in the future. This article reviews the most recent progress and the advances in the field of photocatalytic technologies combined with external electrical field, including the characterization of nano-sized photo(electro)catalysts, the degradation of multi-phase pollutants, and the development of electrical assisted photocatalytic technologies for the potential application on the treatment of organic and inorganic compounds in the wastewaters and waste gases. Innovative oxidation techniques regarding photo(electro)catalytic reactions with and without oxidants are included in this review article.

**Keywords:** advanced oxidation technologies; electrical assisted photocatalysis; external electrical field; reaction mechanisms; organic and inorganic compounds





## 1. Introduction

The application of innovative waste treatment technologies has received great attention in the past for solving the existing environmental pollution problems. Organic and inorganic pollutants such as dyes, chemicals, pharmaceuticals, and heavy metals are the most commonly recalcitrant pollutants in various environments. Many industries including paper, leather, textile, cosmetics, agricultural process, pharmaceutical, food processing, and petrochemical manufacturing could produce a large amount of organic pollutants [1], while other industries including fuel and waste combustion, cement production, iron works, and non-ferrous metallurgy could produce large amounts of inorganic pollutants [2].

Wastes including wastewaters and waste gases can be treated by traditional biological and/or physicochemical processes. Most biological treatments are thought as environmentally friendly as well as relatively inexpensive. However, the application of biological treatment processes is rather limited due to large land space requirements, sensitivity to the chemicals' toxicity, and relatively longer reaction duration. Physicochemical techniques commonly require high equipment cost and commonly demonstrate low effectiveness, especially for the effective removal of dyes and pharmaceuticals [1]. For the past decades, several advanced oxidation technologies (AOTs) have been developed as

effective techniques for decomposing toxic, persistent, and non-degradable contaminants in the wastewaters and waste gases as well [3,4]. AOTs on the basis of in-situ production of strongly active hydroxyl radical (·OH) can non-selectively react with degradable organics and decompose even highly recalcitrant contaminants [5]. Hydroxyl radical is the second highest oxidant followed by fluorine, indicating a high standard reduction potential of $E°$ (·OH/$H_2O$) = 2.80 V (SHE, standard hydrogen electrode) and the rate constants of reactions with several contaminants in the order of $10^6$–$10^{10}$ /M·S [6,7]. Hydroxyl radical has quite short lifetime, approximately a few nano-seconds in the aqueous system, which can be fast self-eliminated from the wastewater treatment system [8].

Typical AOTs include chemical, photochemical, photocatalytic, and photoelectrocatalytic systems. These technologies successfully apply $H_2O_2$ with $UV_C$ radiation (i.e., $H_2O_2$/$UV_C$), ozone and ozone-based processes (i.e., $O_3$, $O_3$/$UV_C$, $O_3$/$H_2O_2$, and $O_3$/$H_2O_2$/$UV_C$), titanium dioxide ($TiO_2$) based processes (i.e., $TiO_2$/UV and $TiO_2$/$H_2O_2$/UV) and Fenton's reaction based methods (i.e., $Fe^{2+}$/$H_2O_2$) and photo-Fenton (i.e., $Fe^{2+}$/$H_2O_2$/UV) [9]. Photocatalytic technique (PCT) uses a semiconductor as photocatalyst (i.e., $TiO_2$) under the light illumination (UV, near-UV, or visible lights) to induce electron and hole pairs ($e_{cb}^-$/$h_{vb}^+$) to degrade organic and inorganic contaminants by producing strong oxidants (·OH and $O_2^-$) at its surface [10]. This technique has prominent advantages including non-toxicity, low cost, no secondary pollution, and high mineralization. However, PC is restricted by its low photonic efficiency due to relatively low rate constants of photocatalytic reactions. The fast recombination of photo-induced $e_{cb}^-$/$h_{vb}^+$ pairs at the photocatalyst surface represents the drawback for potential PC applications [11].

One innovative approach is to integrate the AOTs including electrochemical (EC), photocatalytic (PC), sono-photocatalytic, and photoelectrochemical (PEC) processes in order to enhance the oxidation capability of organic contaminants while compared to the individual process, due to the increased radical formation capability and the enhanced degradation kinetics [7,12,13]. Several previous researchers once utilized photocatalysts as the anode of EC and applied their potential on the photo-anode to drive photo-induced $e_{cb}^-$ away from $h_{vb}^+$, forming more active species to decompose organic pollutants in the wastewaters [14,15]. In the PEC process, organic pollutants degraded by EC are commonly ignored because of low bias that are applied on the photo-anode [16–18]. However, in the PEC system, EC plays the primary contributors for the degradation of organic pollutants. An innovative photo-assisted electrochemical technology is widely developed, whose anode has excellent PC and EC properties. The light irradiation could accelerate the oxidization of intermediates deposited on the surface of anode [19], defer the electrode passivation, and further inhibit the accumulation of intermediates. Therefore, the PEC technique is thought as a promising approach to purify the refractory wastewater while compared to the PC or EC techniques. The enhancement for the degradation efficiency of organic contaminants in the hybrid systems is achieved by the following synergistic effects [17,20,21].

(1) Coupling the processes complement the formation of ·OH radicals via the major reactions of individual processes,

(2) Creating the side reactions mediated by the synergistic effects between individual processes supplement the ·OH radical formation sites and produce other oxidizing radical species [22], and

(3) Combining the mechanisms of photolysis and electrolysis to enhance the mineralization efficiency via the direct PEC of complex intermediates including metal-ligand complexes and carboxylic acid intermediates [3,22–24].

The sono-photocatalytic degradation of organic pollutants occurs mainly due to the synergistic effect of the sonolysis (US) and photocatalysis (UV). The adsorbed dye molecules on the surface of photocatalysts could react with the active radicals formed through the photocatalysis. Subsequently, the desorption of dye molecules from the surface of the photocatalysts by the shock waves generated by the cavitation bubbles would take place, which reduces the interaction probability of dye molecules with the active radicals. On

the other hand, the radicals produced by transient implosion of cavitation bubbles in the vicinity of the photocatalyst particles can cause the effective degradation of dye molecules. Generally, the smaller size and larger surface area of the nano-sized photocatalytic particles increase the active sites, which are more advantageous for sono-photocatalysis [13,25].

## 2. Principles of Photo(electro)catalysis

Electrical assisted photocatalytic and photochemical processes have been investigated for water splitting, wastewater treatment, and waste gas removal in the past. The photo-electrochemical (PEC) process on water splitting was firstly described by Fujishima and Honda (1972) [26] and further investigated from the electrochemical (EC) viewpoint by Bockris et al (1980s) [27], but it was not until the beginning of 21st century as it was used for the treatment of wastewaters in practice [28]. This process utilizes an electrolytic system containing a thin-film photo-anode subjected to the irradiation with the application of a constant bias potential to the anode ($E_{anod}$), a constant cell potential ($E_{cell}$) or a constant anodic current density ($j_{anod}$) [29]. It enhances the extraction of photo-induced $e_{cb}^-$ by the external electrical field, thereby yielding an efficient separation of the $e_{cb}^-/h_{vb}^+$ pairs, thus the reactions in Equations (1)–(5) are effectively inhibited [29,30].

$$\text{photocatalyst} + h\upsilon \rightarrow e_{cb}^- + h_{vb}^+ \tag{1}$$

$$h_{vb}^+ + H_2O \rightarrow \cdot OH + H^+ \tag{2}$$

$$h_{vb}^+ + OH^- \rightarrow \cdot OH \tag{3}$$

$$h_{vb}^+ + R \rightarrow R_{ox}^+ \tag{4}$$

$$e_{cb}^- + O_2 \rightarrow \cdot O_2^- \tag{5}$$

Retarding the recombination of photo-induced $e_{cb}^-/h_{vb}^+$ could accelerate the rate of redox reactions and thus upgrades the photocatalytic activity of the photo-anode with the generation of higher amounts of photo-induced $h_{vb}^+$ and the acceleration of organic degradation while compared to the traditional PC technology. As a result, the lifetime of $h_{vb}^+$ is efficiently prolonged and have more opportunities to either directly decompose organic pollutants adsorbed on the surface of photo-anode or indirectly destroy them with more amounts of $\cdot OH$ formed by Equation (2) [30]. In the PEC process, the photocatalysts can be easily recovered after usage and recycled for the consecutive treatments. A catalytic semiconductor (M) submitted to an anodic potential can decompose organic pollutants by electrochemical oxidation (ECO), since it can oxidize water to hydroxyl radical and form M($\cdot OH$) (Equation (6)) [29,30]

$$M + H_2O \rightarrow M(\cdot OH) + H^+ + e_{cb}^- \tag{6}$$

Photo-induced $h_{vb}^+$ is a strong oxidizing species, whereas the promoted $e_{cb}^-$ is a potential reducing species (Equation (1)) mineralizing organic contaminants, though no solid evidences support the formation of $\cdot OH$ from $h_{vb}^+$ (Equations (2) and (3)). The photo-induced $e_{cb}^-$ can react with the adsorbed $O_2$ and form the superoxide radical $O_2^-$ (Equation (5)) [29–31].

The photoelectrocatalytic (PEC) technique inherits the advantage of photocatalysis while enhancing the radical formation capacity with the combination of photocatalytic and electrolytic reactions [20]. Photocatalysis is one of the extensively explored AOPs for mineralizing multi-phase pollutants. Upon the irradiation of UV, near-UV, and visible lights, semiconductor-based photocatalysts such as zinc oxide (ZnO), titanium dioxide ($TiO_2$), vanadium oxide ($V_2O_5$), tungsten trioxide ($WO_3$), and bismuth vanadate ($BiVO_4$) could produce reactive oxygen species ($\cdot OH$ and $O_2^-$) [32].

As the incident light with energy ($h\upsilon$) greater than the band-gap energy is irradiated on the photocatalysts, the photo-induced electron ($e_{cb}^-$) can be excited from the valence band (VB) to the conduction band (CB), leaving behind a positive vacancy hole ($h_{vb}^+$)

in the VB and effectively generating ·OH radicals through the oxidation of $H_2O/OH^-$ (Equations (2) and (3)). However, the radical formation capacity of the photocatalysts is rapidly reduced due to the fast recombination of photo-induced $e_{cb}^-/h_{vb}^+$ pairs since the lifetime of $e_{cb}^-/h_{vb}^+$ pairs is in the order of nanoseconds. Nevertheless, the recombination of photo-induced $e_{cb}^-/h_{vb}^+$ pairs can be retarded by the external electrical potential [33]. In the PEC process, photocatalysts can be coated on the conductive supporting matrix (called as photo-anode). As the photo-induced $e_{cb}^-$ are generated upon the irradiation of lights with suitable wavelength, the external applied positive potential to the photo-anode forces the transfer of photo-induced $e_{cb}^-$ to the cathode. This phenomenon favors $h_{vb}^+$ to produce more radicals through the reaction with $H_2O/OH^-$ (Equations (2) and (3)) [34,35].

The degradation efficiency of organic pollutants (R) for the PEC technique could be further enhanced by the direct oxidation of organic pollutants depending upon the direct contact to $h_{vb}^+$ as shown in Equation (4). Furthermore, superoxide anion can be formed as a side reaction product between the electro-excited electrons and the oxygen molecules (Equation (5)). Other weaker reactive oxygen species (ROS) such as hydroperoxyl radical ($HO_2·$) and hydrogen peroxide ($H_2O_2$) can be also produced by Equations (7) and (8).

$$·O_2^- + H^+ \rightarrow HO_2· \tag{7}$$

$$2HO_2· \rightarrow H_2O_2 + O_2 \tag{8}$$

Nevertheless, the photo-promoted $e_{cb}^-$ is an unstable species with excited state and tends to return back to its ground state either with adsorbed ·OH radical in Equation (9) or preeminently by recombining with the unreacted $h_{vb}^+$ as shown in Equation (10).

$$e_{cb}^- + ·OH \rightarrow OH^- \tag{9}$$

$$e_{cb}^- + h_{vb}^+ \rightarrow \text{photocatalyst} + \text{heat} \tag{10}$$

The application of external electrical field coupling with photocatalysis could significantly enhance the degradation efficiency of organic pollutants. It is mainly attributed to retard the recombination of $e_{cb}^-/h_{vb}^+$ pairs and promote the formation of radicals simultaneously in both photocatalytic and electrochemical processes. For photo-anodes, $e_{cb}^-$ are transferred to the external circuit and are concentrated in the cathode, while $h_{vb}^+$ are accumulated in the anode, which guarantees a significant decrease in the recombination rate of $e_{cb}^-/h_{vb}^+$ pairs. Such reactive radicals and oxidative species formed on the cathode and anode, respectively, can react with organic contaminants in the wastewaters. Figures 1 and 2 illustrate the mechanisms for the photoelectrocatalytic degradation of organic contaminants for using the n-type or p-type semiconductors [36,37].

The light absorption of photocatalysts and the separation capability of photo-induced $e_{cb}^-/h_{vb}^+$ are two crucial operating factors influential to the photo-activity of organic contaminants. Upon the absorption of photon by $TiO_2$, photo-induced $e_{cb}^-/h_{vb}^+$ need to be separated and transferred to the surface of $TiO_2$, and further react with corresponding water molecules or sacrificial agents (e.g., KOH electrolyte). The $e_{cb}^-$ may migrate to the nano-cylinder of carbon nanotubes (CNTs) causing a reduction in the chance of the recombination of $e_{cb}^-/h_{vb}^+$. Upon light irradiation, the electron ($e_{cb}^-$) are excited from the valence band (VB) to the conduction band (CB), leaving positive $h_{vb}^+$ in the VB [38].

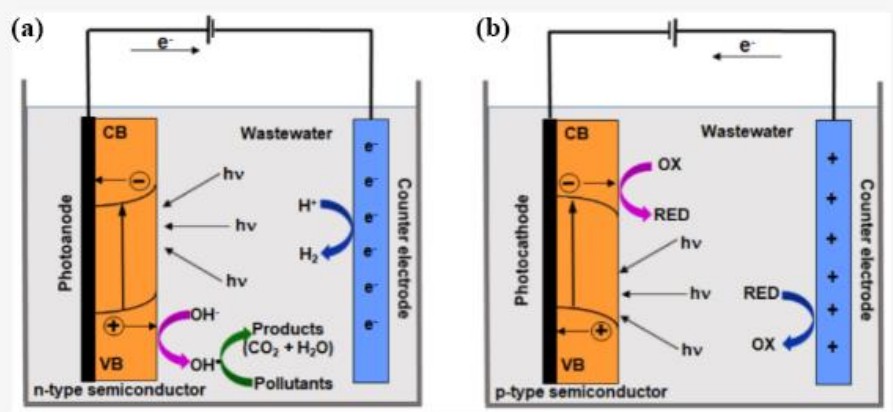

**Figure 1.** Mechanisms of photoelectrocatalysis and the main reactions for (**a**) n-type and (**b**) p-type semiconductors [39].

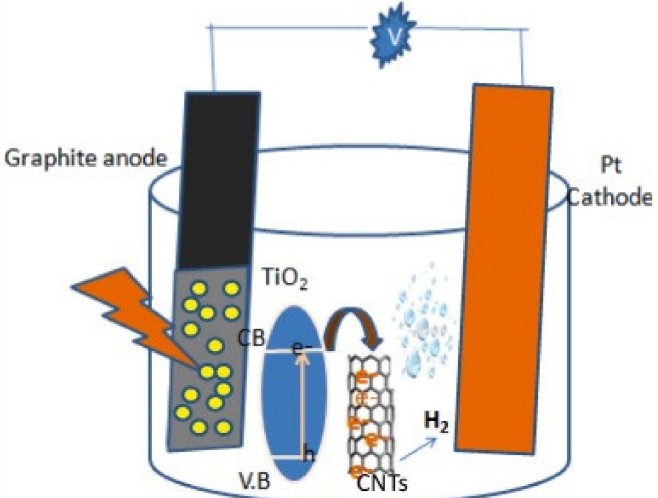

**Figure 2.** Mechanisms of $H_2$ PEC generation using CNTs @ TNRs in the presence of 0.9 M KOH [36].

## 3. Mechanisms of Photo(electro)catalytic Reactions

### 3.1. Chemical Kinetics of Photo(electro)catalytic Reactions

The chemical kinetics of photo(electro)catalytic reactions have been developed to describe the degradation of pollutants in the aqueous phase. A Langmuir-Hinshelwood (L-H) kinetic model can be applied to successfully evaluate the heterogeneous photocatalytic reactions in either liquid-solid or gas-solid interfaces. The kinetics of organic pollutant oxidation can be simulated by the L-H kinetic model as shown below [40],

$$\ln\left(\frac{C_0}{C_t}\right) = k_r\mathrm{Kt} = K_{\mathrm{app}}t \qquad (11)$$

A pseudo-first-order kinetic model for the photo-degradation of cyanide in a nano-sized photocatalytic reaction system is depicted in Figure 3. Experimental data are highly correlated with the pseudo-first-order kinetic model with a determination coefficient ($R^2$) of 0.98. The rate constant ($K_{\mathrm{app}}$) of photocatalytic reaction was obtained from the slope of the plot between $\ln(C_0/C)$ versus t (irradiation time) as illustrated in Figure 3. Detail data regarding the estimated kinetic parameters are summarized and shown in Table 1. It is observed that the $K_{\mathrm{app}}$ value of FeTCPP-S-$TiO_2$@rGO in a solid-state photocatalytic system is approximately 1.7 times higher than that in a suspended photocatalytic system. Furthermore, the reaction rate of photocatalytic activity by using the nano-sized photocatalysts was nearly 10.8 times higher than that of bare $TiO_2$ [41]. Similarly, for the photocatalytic

oxidation of cyanide, ampicillin, and oxytetracycline, apparent rate constants ranging from 0.0034 to 0.027 /min have been depicted in the literature [42].

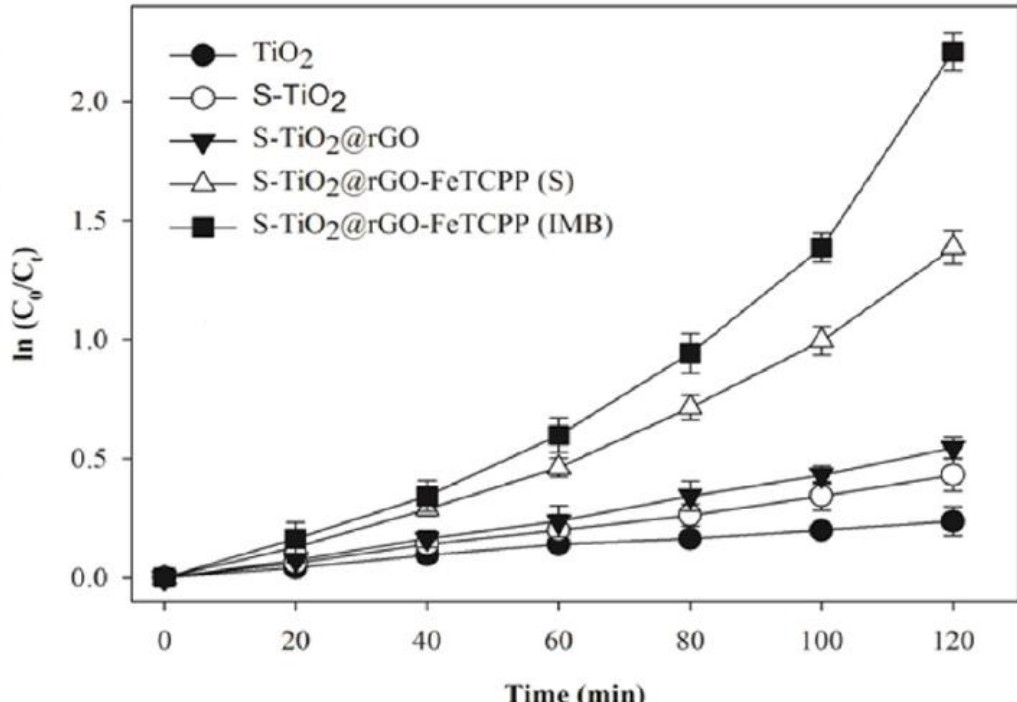

**Figure 3.** Pseudo-first-order kinetic plot for degradation of cyanide by Fe-TCPP-S-TiO$_2$@rGO photocatalytic system under visible light [41].

**Table 1.** Estimated kinetic parameters in the photocatalytic degradation of cyanide using FeTCPP-S-TiO$_2$@rGO nanocomposite system [41].

| Photocatalysts | Cyanide Degradation (%) | K$_{app}$ | R$_2$ |
|---|---|---|---|
| TiO$_2$ | 21%(3.25) | 0.0018 | 0.9746 |
| S-TiO$_2$ | 35%(3.51) | 0.0035 | 0.9938 |
| S-TiO$_2$@rGO | 41%(4.04) | 0.0041 | 0.9929 |
| FeTCPP-S-TiO$_2$@rGO(SUS) | 75%(5.03) | 0.0113 | 0.9843 |
| FeTCPP-S-TiO$_2$@rGO(IMB) | 91%3.75 | 0.0196 | 0.9880 |

Figure 4 depicts the kinetic plot of $-\ln (C_t/C_o)$ as a function of irradiation time (t) for the photo-degradation of typical dyes (i.e., membrane orange (MO), crystal violet (CV), membrane blue (MB), and rhodamine B (RhB)) [43]. It confirms that the photocatalytic degradation phenomenon fitted quite well with the pseudo-first-order kinetics. The rate constants (k) of 0.007, 0.008, 0.009, and 0.012 /min are determined for the dyes MO, MB, CV, and RhB, respectively. It indicates that the NiO nano-belts can deliver a higher rate constant for RhB and a lower rate constant for MO. Furthermore, the determination coefficients (R$^2$) of MO, MB, CV, and RhB are 0.913, 0.923, 0.961, and 0.965, respectively. It illustrates better response of NiO nano-belts for the photo-degradation of RhB and CV than that of MO and MB.

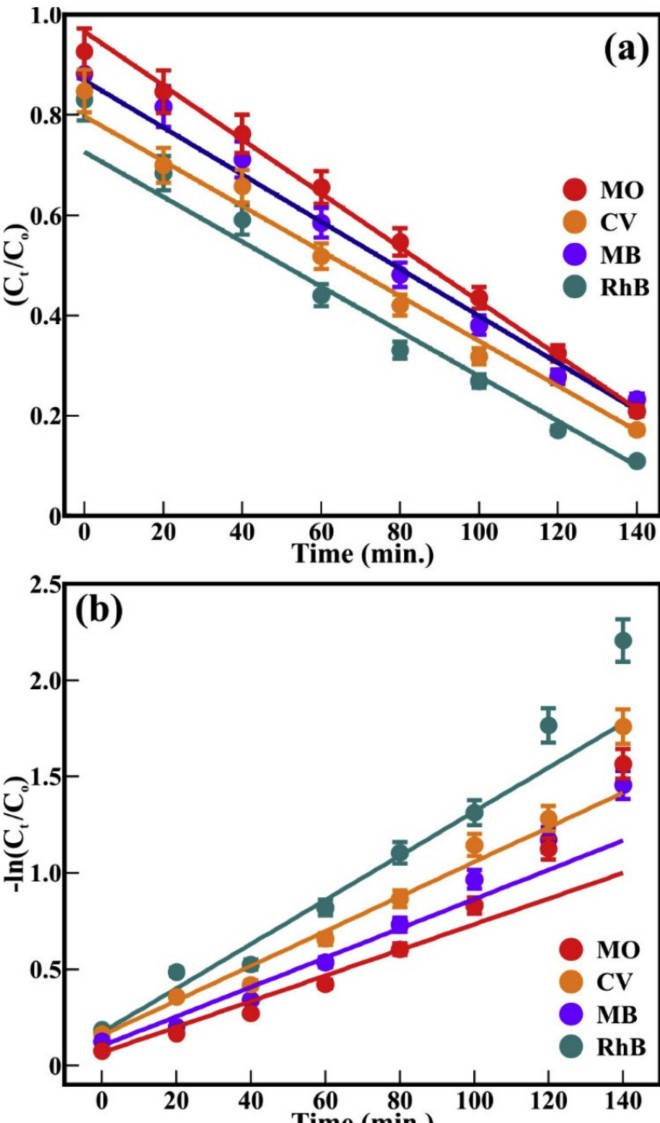

**Figure 4.** (**a**) Kinetic curves and (**b**) corresponding pseudo-first-order kinetic plot [43].

### 3.2. Model Simulation of Photo(electro)catalytic Reactions

　　Previous literature once reported that traditional photocatalysts occasionally fail to achieve satisfactory performance for the photocatalytic reaction process [44]. Various measures have been applied to successfully prepare optimized photocatalysts via doping process. However, it is commonly a tedious and fussy process with time consumption and experimental errors to explore the effective resolution for the optimization of photocatalysts. In order to save time and reduce experimental errors, the density functional theory (DFT) has been gradually utilized to establish the electronic structure and to characterize the chemical property of photocatalysts. Theoretically understanding the physicochemical properties of photocatalysts can be achieved by DFT, and thus to facilitate the optimization process. The DFT is not only indispensable to establish the electronic structure and the chemical property of photocatalysts, but also provides valuable guidance to understand the mechanisms involved in the photocatalytic reaction processes [45–47].

　　Firstly, the stability analysis can allow us to establish the starting vanadium-titanium dimeric structure. Three potential configurations of vanadium dimeric structures are illustrated in Figure 5, differing only in the orientation of the V-O(2)-V bond and the missing surface oxygen atoms. Amongst the three dimeric structures mentioned above, structure A is chosen as a structure used for the subsequent calculations due to its lowest formation energy and the smallest change in the interlayer distance with respect to structures B and C.

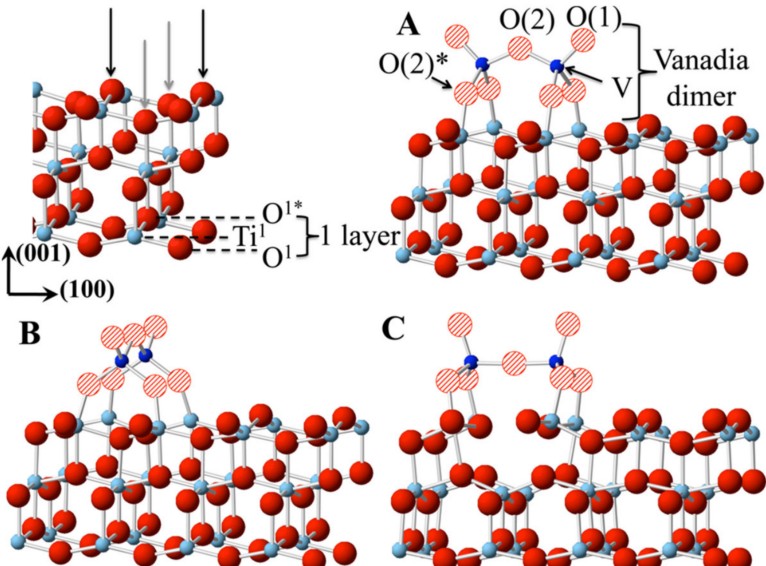

**Figure 5.** Clean surface and three possible configurations of the vanadium dimers. Dark blue atoms are V, red atoms are O from the support, striped red atoms are O atoms from vanadium dimer, and light blue atoms are Ti. Black arrows indicate the O atoms that are missing in structures (**A**,**B**), while gray arrows indicate the O atoms missing in structure (**C**) [47].

With the purpose of having better representation of a commercial photocatalyst for selective catalytic reduction (SCR) ($V_2O_{5(<2\ wt\%)}$-$TiO_2$) with its vanadium content below monolayer regime, the active phase is modeled as a vanadium dimer supported on a 3-layer $TiO_2$ slab [23]. This model lets us to ascertain the active sites participating in the photocatalytic activity of vanadiyl oxygen (V = O(1)) and bridging O (V-O(2)-V). The amount of water vapor exists in the flue gas ($\approx$10%) and the stabilization effect of water molecules adsorbed on the $TiO_2$ surface allow us to explore the thermodynamic stability of different hydrated and hydroxylated surfaces as a function of pressure and temperature. Concurring with previous works regarding the experiments and theoretical modeling, water could exist in a molecular layer or in a dissociative fashion depending upon the coverage of water layer over the $TiO_2$ surface, which is highly temperature dependent. A dehydration process occurs at the reaction temperatures above 390 K, resulting in a water free surface of the photocatalysts.

The activity of vanadium dimer is further analyzed by the adsorption energies of Hg, HgCl, HCl, $H_2O$ and their combinations, which are mostly likely involved in the oxidation mechanism of Hg. These adsorption energies are highly correlated with Bader charge analysis and partial density of state (PDOS) of vanadium dimer for the increased understanding of the changes in the electronic structure upon dehydration. It is shown that the adsorbed water molecules act as a Lewis base, donating electrons to the support, which increases the negative charge on the oxygen atoms of vanadium dimer. The increase in charge leads to higher activity of oxygen atoms, resulting in stronger vanadium-titanium interactions. The photocatalysts' surfaces with high water coverage show high activity with HgCl, exhibiting the highest adsorption energy and followed by HCl. The adsorption energies of Hg imply a weak interaction with the vanadium dimer occurring only on the surfaces with high water coverage. Moreover, ab initio thermodynamic calculations are carried out to consider the effects of temperature and entropy loss on the adsorption energies of Hg, HgCl, HCl, $H_2O$ and their combinations. The adsorption energies of the chemical species commonly decrease as reaction temperature increases, becoming energetically unfavorable at T > 300 K.

A mechanism applied to describe the photo-oxidation of Hg has been proposed on the basis of adsorption energies at 300 K. This mechanism involves the adsorption of HCl and HgCl, following the L-H adsorption mechanism. The oxidation of Hg to $HgCl_2$ is presented

as a cycle without poisoning the surface during the interaction of HgCl$_2$ and HCl, since the original photocatalysts involve the first through the last steps of the reactions. During the process of HgCl$_2$ formation, some steps involve the adsorption and dissociation of HCl, the formation and adsorption of HgCl, and the formation and desorption of HgCl$_2$ from the surface of photocatalysts. The active vanadium phase (i.e., V$_2$O$_5$) is responsible for the objective reduction reactions for reducing NO$_X$ to N$_2$ and other undesired oxidation reactions for oxidizing SO$_2$ to SO$_3$. The impacts of other flue gas components, such as SO$_2$, SO$_3$, NO, and NO$_2$, on mercury oxidation across the SCR unit should be investigated in the future work [47].

Results obtained from DFT aid in the design of improved thermos-catalysts for mercury oxidation to simultaneously maintain optimal performance of NO$_x$ reduction. DFT calculations have been performed to determine and compare the adsorption of O$_2$ and NO on the surface of TiO$_2$ and OVs-TiO$_2$, respectively [48]. As illustrated in Figure 6a1–d2, molecules NO and O$_2$ are relatively difficult to be adsorbed on the surface of pristine TiO$_2$, while they are easily adsorbed on the surface of OVs-TiO$_2$. Particularly, the distances between molecules NO and O$_2$ and the surface of pristine TiO$_2$ are 1.525 and 2.417 Å, respectively (see Table 2). In comparison, the distances between molecules NO and O$_2$ on the surface of OVs-TiO$_2$ both decrease to 0 Å. Moreover, the adsorption energies of NO and O$_2$ decrease from 0.836 and 1.020 eV on the surface of TiO$_2$ to −1.941 and −3.776 eV on the surface of OVs-TiO$_2$, respectively. Table 2 shows that both band lengths of NO and O$_2$ are respectively lengthened from 1.165 and 1.236 Å(absorbed on the surface of TiO$_2$) to 1.301 and 1.395 Å(absorbed on the surface of OVs-TiO$_2$). Thus, it concludes that OVs could capture NO molecule and activate N-O band, which favors the photocatalytic reduction of NO. It is of importance to note that OVs could activate O$_2$ molecule to create reactive oxygen species (ROS) which play crucial roles in photocatalytic NO reduction process.

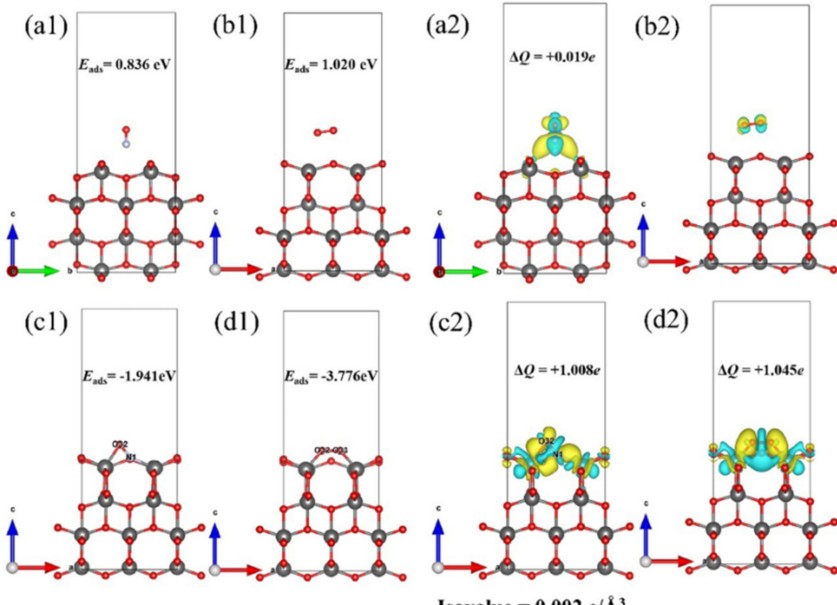

**Figure 6.** Optimal structures for the absorption of molecules NO and O$_2$ on pristine TiO$_2$ (**a1,b1**) and OVs-TiO$_2$ (**c1,d1**) and the charge density difference of optimized NO and O$_2$ absorbed on pristine TiO$_2$ (**a2,b2**) and OVs-TiO$_2$ (**c2,d2**) [48].

**Table 2.** The distance between molecules NO and $O_2$ and the surface of $TiO_2$ and OVs-$TiO_2$ based on the DFT calculations, respectively, and their corresponding band lengths of molecules NO and $O_2$ adsorbed on the surface of $TiO_2$ and OVs-$TiO_2$ [48].

| Molecules | Distances (Å) | | Band Lengths (Å) | |
|---|---|---|---|---|
| | NO | $O_2$ | NO | $O_2$ |
| $TiO_2$ | 1.525 | 2.417 | 1.165 | 1.236 |
| OVs-$TiO_2$ | 0.0 | 0.0 | 1.301 | 1.395 |

Additionally, the charge density difference of molecules $O_2$ and NO adsorbed on the surface of $TiO_2$ and OVs-$TiO_2$ are illustrated in Figure 6a2–d2, respectively. Herein, light blue represents the charge loss, while light yellow stands for the charge accumulation. Based on the Bade charge analysis, there is almost no electrons transferring from the surface of $TiO_2$ to the free molecules NO and $O_2$. Oppositely, electrons could be transferred from the surface of OVs-$TiO_2$ to the adsorbed NO (1.008 e) and $O_2$ (1.045 e) molecules, respectively, which inferred the mechanism of chemisorption. As a result, OVs is beneficial for both surface adsorption and thermo-catalytic reaction, leading to the enhanced thermos-catalytic activity of OVs-$TiO_2$-$N_2$. Based on the DFT calculations, the facile formation of OVs is mainly attributed to the intrinsic carbon-doping effect. The OVs-$TiO_2$-$N_2$ exhibits significantly improved thermos-catalytic activity, which is approximately four times higher than that of pristine $TiO_2$. Both experimental and theoretical results demonstrate that the thermo-catalytic reduction of NO can be enhanced mainly due to the curial roles of OVs [48].

Deep insights into the $O_2^-$ formation on the NiO surface play a crucial role for the degradation of dyes. It is gained through the state-of-the-art calculations based on the DFT calculation to investigate the adsorption reactions between $O_2$ and NiO(110) surface [43]. Prior to conducting the adsorption characterization of $O_2$, the electronic structure of NiO is initially determined (Figure 7a,b), predicting a bandgap of 3.76 eV for bulk NiO. The partial density of states (PDOS) reveals that O-p states dominate the valence band (VB), whereas Ni-d states dominate the edge of conduction band (CB). The lowest energy of $O_2$ adsorbed geometries on the surface of Ni(110) is illustrated in Figure 7c,d. The adsorption energies for $O_2$ adsorbed side-on and end-on at Ni sites estimated by DFT are −3.59 and −1.62 eV, respectively. It obviously indicates that the side-on $O_2$ configuration is energetically more favored than the end-on configuration. The O-O bond distances for $O_2$ bound in the side-on and end-on geometries are predicted as 1.360 and 1.278 Å, respectively, indicating elongation relative to gas-phase $O_2$ molecule (1.24 Å). Bader population indicates that $O_2$ molecule gained −0.67 and 0.40 $e^-$ as bound in the side-on and end-on geometries, respectively, from the interacting Ni sites on the surface, thus resulting in the formation of $O_2^-$ radicals. Atomic-level insights into the electron density redistribution within the $O_2$/NiO(110) system is obtained from the differential charge-density difference (Δρ) iso-surfaces analysis that is obtained by subtracting from the electron density of the total $O_2$/NiO(110) systems, the electron densities of both naked NiO(110) surface and isolated $O_2$ molecule (i.e., Δρ = ρ$O_2$/NiO(110) − ρNiO(110) − ρ$O_2$). As depicted in Figure 7e,f, the iso-surface contours reveal the chemisorption of $O_2$ molecule, characterized by electron density accumulation (blue contours) within the newly formed Ni-O bonds and on $O_2$ molecules. The newly formed $O_2^-$ radicals on the surface of NiO(110) are expected to react with RhB dye and thus facilitate their degradation [43].

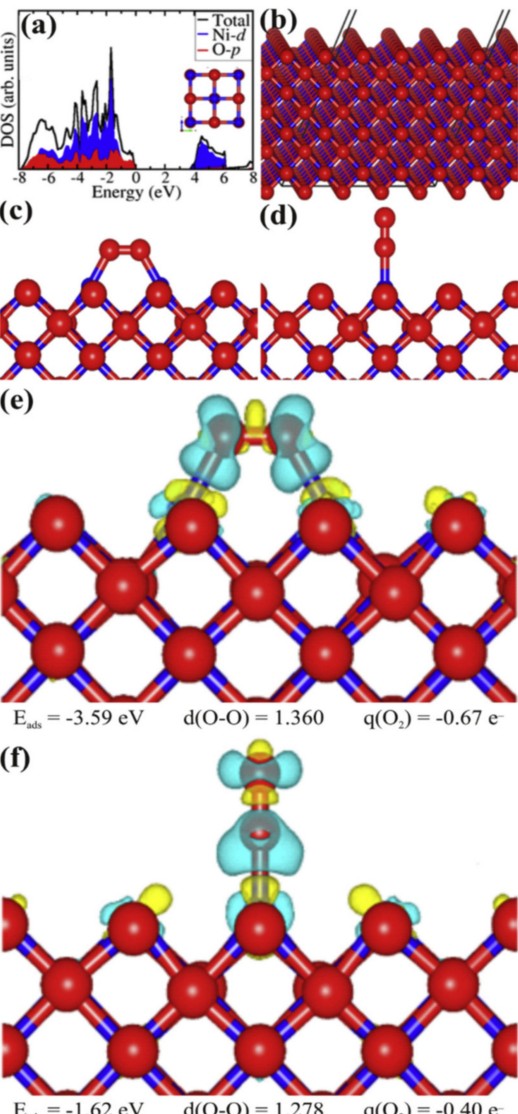

**Figure 7.** (**a**) PDOS of rock salt NiO (inset), (**b**) optimized surface structure of NiO(110) surface. The relaxed adsorption structures of $O_2$ adsorbed in (**c**) side-on and (**d**) end-on configurations, with the corresponding differential charge density iso-surface contours (**e**,**f**), where green and yellow contours indicate electron density increase and decrease by 0.003 $e/Å^3$, respectively. Blue and red balls represent Ni and O atoms, respectively [43].

## 4. Types of Photo(electro)catalysts

The degradation efficiency of environmental contaminants is highly related to the semiconductors chosen as photocatalysts by their intrinsic photocatalytic properties. Commonly, the photocatalysts are mostly metallic oxides with an appropriate energy gaps between the fulfilled valence band (VD) and the empty conduction band (CB) to produce photo-induced $e_{cb}^-/h_{vb}^+$ pairs upon light irradiation. Several semiconductors used as potential photocatalysts in the photoelectrocatalytic technique for the treatment of wastewaters and waste gases are introduced and described in detail shown below.

### 4.1. Metallic Oxides

Previous studies have devoted to effectively remove organic contaminants from wastewaters and waste gases by using photo(electro)catalysts for the innovative photoelectrocatalytic (PEC) techniques. The removal efficiency of PEC technique on the remediation of organic pollutants depends upon the usage of semiconductors as photo-anodes according to their intrinsic PEC properties. They are generally metallic oxides with appropriate

band gap energies ($E_{bg}$) between the fulfilled valence band (VB) and the empty conduction band (CB) to produce the photo-induced $e_{cb}^-/h_{vb}^+$ pairs upon light irradiation. Different photo(electro)catalysts have been applied in the PEC technique for the effective treatment of organic or inorganic contaminants in the wastewaters and waste gases. The photo(electro)catalysts are commercially available or non-available semiconductors.

To date, the most commonly used commercial available photo(electro)catalyst is titanium dioxide ($TiO_2$) which is an n-type semiconductor with three crystalline structures, namely anatase, rutile, and brookite. The energy band ($E_{bg}$) of $TiO_2$ is slightly superior to 3.0 eV, with slight differences between the crystalline phases. The $E_{bg}$ of anatase, rutile, and brookite are 3.23, 3.02, and 3.14 eV, respectively. Among them, anatase $TiO_2$ with the highest $E_{bg}$ is the most active phase in PC and PEC techniques upon UV irradiation and produces the photo-induced $e_{cb}^-/h_{vb}^+$ pairs. Degussa P-25 is the most commonly commercial available semiconductor which has been widely applied in many environmental fields, particularly for the photo(electro)catalysts (PC and PEC) and the thermocatalysts (SCR) due to its non-toxic, non-reactive, and luminous properties. Anatase $TiO_2$ is the most utilized metallic oxide photocatalyst used for the applications of environmental remediation and pollution treatment [49].

Other metallic oxide, zinc oxide (ZnO), extensively and commonly exists in the natural crust and is considerably cheaper than $TiO_2$. It is also an environmental-friendly semiconductor due to its innocuous character for the health of human beings. ZnO has two typical crystalline structures, namely hexagonal wurtzite and cubic zinc blende. The $E_{bg}$ value of wurtzite is 3.4 eV with high electrochemical stability and highly available applications under the solar irradiation [50]. The thin films of ZnO are basically transparent, which improves the penetration of incident lights into the semiconductor and thus enhances the formation of photo-induced $e_{cb}^-/h_{vb}^+$ pairs.

Tungsten oxide ($WO_3$) is another n-type semiconductor, whose crystalline structures are moderately temperature dependent. The $E_{bg}$ value of $WO_3$ is in the range of 2.5–2.7 eV, which are appreciably lower than $TiO_2$. As a result, $WO_3$ presents better performance under the irradiation of visible light due to its easier formation of photo-induced $e_{cb}^-/h_{vb}^+$ [51]. The first usage of $WO_3$ photo-anode for wastewater treatment has been proposed by Hepei and Luo (2001) [52]. The main drawback of $WO_3$ is that it is not as innocuous as $TiO_2$, since it is a hazardous, toxic, and irritant semiconductor [53].

In addition to ZnO, $TiO_2$, and $WO_3$, other metallic oxide semiconductors have been used as photo-anodes in the photoelectrocatalytic techniques searching for low cost, high efficient, and chemically stable photocatalysts to have electrical current with the capability to obtain larger energy from solar spectrum to photo-induced $e_{cb}^-/h_{vb}^+$ pairs. However, the overall performance of these semiconductors cannot be easily compared with that found with classical $TiO_2$ due to different PEC systems and operating parameters used for environmental pollution treatment. For example, the hematite-$Fe_2O_3$ has been considered as one of the potential candidates mainly due to its low cost, high chemical stability, and high absorbance. Low $E_{bg}$ value of 2.2 eV allows the application of visible light energy [54,55]. Other potential promising semiconductor is MnO with an even low $E_{bg}$ value of 1.3 eV to be an interesting functional material with low cost, large surface, electrochemical stability [55], and innocuous character [56,57]. Further combining MnO with polyaniline can demonstrate the enhancement of PEC properties because of the interaction with the polyaniline's $E_{bg}$ of 2.8 eV, which could retard the recombination of photo-induced $e_{cb}^-/h_{vb}^+$ [57]. Additionally, $SnO_2$ is a chemically and thermally stable semiconductor with wide $E_{bg}$ values of 3.5–3.8 eV, which is less suitable for using as a photocatalyst [58]. However, the non-conductive $SnO_2$ can be easily modified by doping it with Sb to considerably improve its electrical conductivity, the doped Sb-$SnO_2$ is then employed as photocatalyst that presents low electrochemical stability with short lifetime upon current applications [59]. Another semiconductor β-$PbO_2$ can be utilized as photocatalyst owning to its relatively small $E_{bg}$ value of 1.4 eV [60], but β-$PbO_2$ cannot be used in acidic solutions, because it would leach toxic Pb to the medium under acidic conditions.

In contrast, some stable bismuth-vanadium materials such as BiVO$_4$ with the E$_{bg}$ of 2.5 eV and Bi$_2$WO$_5$ with the E$_{bg}$ of 2.8 eV [61] under the visible light irradiation and BiPO$_6$ with the E$_{bg}$ of 3.8 eV [62] under the UV irradiation, which show as excellent effectiveness and good performance for the PEC treatment of organic contaminants.

*4.2. Doped/Modified Metallic Oxides*

The performance of PEC techniques mainly depend upon the following three influential factors: (1) the properties of incident light absorption, (2) the rate of redox reactions occurred on the surface of the photo(electro)catalysts, and (3) the recombination rate of photo-induced e$_{cb}^-$/h$_{vb}^+$ pairs. In order to enhance the PEC activity of the photo(electro)catalysts, various optional measures have been applied including: (1) reconstructing the structure of metallic oxides to nanotubes (NTs), nanoarrays (NAs), nanobelts (NBs), or nanorods (NRs), (2) doping the photo(electro)catalysts with metals such as Cr, Cu, and Fe, or non-metals such as B and N, (3) synthesizing the composites with metals such as Pd, Ag, and Au; metallic oxides such as SiO$_2$, WO$_3$, and Sb$_2$S$_3$; and carbon materials such as carbon fabric and graphene, and (4) developing the innovative titanium containing materials such as TiNbO$_5$ and the cubic double-perovskite like CaCu$_3$Ti$_4$O$_{12}$.

The traditional photocatalysts such as TiO$_2$ doping with metals or non-metals would introduce newly built VB and CB related to the impurity energy levels that enhance the degradation of organic contaminants by the PEC technique while compared to the photocatalysts with no doping. Kerkez et al. (2014) [63] synthesized TiO$_2$ nanorod array film deposited onto fluorine-doped tin oxide (FTO) further doped with CuO up to the Cu$^{2+}$ molar ratio of 0.26%. The E$_{bg}$ values of these doped photocatalysts gradually decrease with rising dopant proportion, from 3.1 eV for the bare TiO$_2$/FTO to 2.6 eV for the higher doped photo-composites, which allows the photocatalytic reaction system to effectively operate under the irradiation of visible light, improving the PEC treatment of organic contaminants [63]. It can be explained by the transference of photo-induced e$_{cb}^-$ in TiO$_2$/CuO to less energetic CB of CuO to reduce divalent Cu$^{2+}$ to univalent Cu$^+$ which can be achieved with O$_2$ or at the photo-anode. Another potential strategy for enhancing the photocatalytic activity of TiO$_2$ is to prepare composites doped with noble metals and other semiconductors. The PEC technique can be enhanced by decorating TiO$_2$ NTs with Pd, Ag, and Au nanoparticles [17,18,21]. This is mainly attributed to the lower work function of noble metals than the electron affinity of TiO$_2$ NTs that originates a Schottky barrier potential in the interfaces. It makes energetically favorable electron transfer from the CB of semiconductors to the metals and significantly retards the recombination of photo-induced e$_{cb}^-$/h$_{vb}^+$.

## 5. Physicochemical Characteristics of Photo(electro)catalysts

*5.1. Chemical Properties of Photo(electro)catalysts*

Although TiO$_2$ can photo-chemically possess an appropriate CB for the reduction of water and forms hydroxide radicals (·OH), it commonly shows poor water splitting activity by using PEC with the irradiation of visible light and worse photocatalytic performance owing to its wide band gap (E$_{bg}$). However, the band gap of Bi$_2$WO$_6$ is relatively small, but it presents relatively poor PEC water splitting activity and photo-efficiency due to its impropriate CB level for the reduction of water splitting and rapid recombination of photo-induced e$_{cb}^-$/h$_{vb}^+$ [46]. Hence, the core-level X-ray photoelectron spectroscopy (XPS) and VB spectra are measured to identify the band alignment. The valence band maxima (VBM) for TiO$_2$ and Bi$_2$WO$_6$ are determined by VB with a liner extrapolation measure. As depicted in Figure 8a–d, the VBMs of TiO$_2$ and Bi$_2$WO$_6$ are determined as 2.46 and 2.05 eV [64], respectively, concurring with the previously reported data [65,66]. Based on the experimental data of the heterojunction, the CB offsets between TiO$_2$ and Bi$_2$WO$_6$ in TiO$_2$/Bi$_2$WO$_6$ heterojunction can be determined by Equations (12) and (13) [65].

$$\Delta E_{v(TiO_2/Bi_2WO_6)} = \left(E_{Bi_{4f7/2}} - E_{VBM}\right)_{Bi_2WO_6} - \left(E_{Ti_{2p3/2}} - E_{VBM}\right)_{TiO_2} - \left(E_{\boldsymbol{Bi_{4f7/2}}} - E_{Ti_{2p3/2}}\right)_{TiO_2/Bi_2WO_6} \tag{12}$$

$$\Delta E_{C(TiO_2/Bi_2WO_6)} = E_{gBi_2WO_6} - E_{gTiO_2} + \Delta E_{v(TiO_2/Bi_2WO_6)} \tag{13}$$

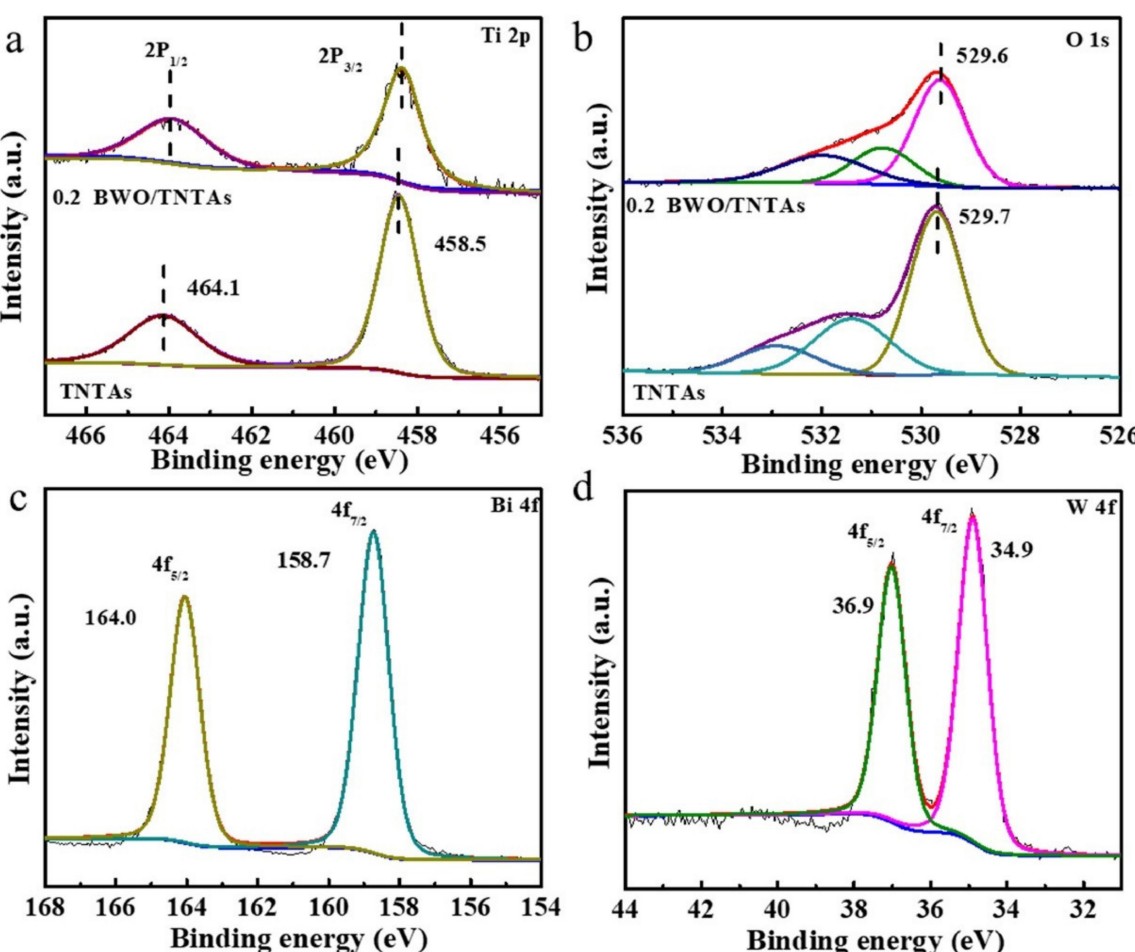

**Figure 8.** XPS spectra of 0.2 BWO/TNTAs and pristine TNTAs sample: (**a**) Ti2p, (**b**) O1s, (**c**) Bi4f, and (**d**) W4f [61].

As illustrated in Figure 8, the values of $\Delta E_V$ and $\Delta E_C$ are determined as 0.71 and 0.38 eV, respectively, by using Equations (12) and (13). These results indicate that the CB position of $Bi_2WO_6$ is higher than that of $TiO_2$, and the VB position of $Bi_2WO_6$ is lower than that of $TiO_2$. It apparently shows that a type II band alignment occurs in the interface of $TiO_2$ and $Bi_2WO_6$. The type II band alignment can transfer electrons from the CB of $Bi_2WO_6$ to that of $TiO_2$.

A schematic diagram of band energies and charge transfer in the $TiO_2/Bi_2WO_6/Au$ system is proposed and illustrated in Figure 9. The photo-induced electron ($e_{cb}^-$) moves to the target CB of $TiO_2$ from that of $Bi_2WO_6$, realizing the effective separation of charge carriers via visible light driven PEC water splitting activity. Moreover, for the fabricated hierarchical $TiO_2/Bi_2WO_6/Au$ NRAs, the hot electrons ($e_{cb}^-$) in the Au NPs can be excited by surface plasma resonance (SPR) to the CB of $Bi_2WO_6$, then migrate to the CB of $TiO_2$ and participate in the following water reduction process, leading to drastically enhanced PEC water splitting activity while compared to the hierarchical $Bi_2WO_6$ NRAs. Oppositely, the electrons ($e_{cb}^-$) photo-induced by light irradiation and SPR can be effectively captured by oxygen to form superoxide radicals ($O_2^-$), and further form hydrogen peroxide ($H_2O_2$), which further reacts with electrons ($e_{cb}^-$) to form hydroxyl radicals ($\cdot OH$). Moreover, the holes ($h_{vb}^+$) on the VB of $Bi_2WO_6$ could oxidize the adsorbed water molecules ($H_2O$) and form hydroxyl radicals ($\cdot OH$) [67].

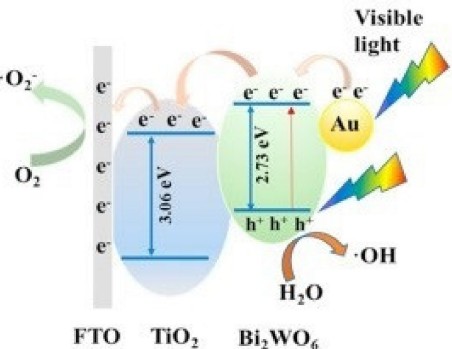

**Figure 9.** Schematic diagram of band energies and charge transfer in $TiO_2/Bi_2WO_6/Au$ system [67].

The effects of metal doping on the $TiO_2$ film, the crystallite size, the lattice distortion, and the cell parameters of the catalysts are investigated by applying X-ray diffraction (XRD) core-shell heterojunction architecture patterns. The lattice parameters and the cell volume can be determined by using Rietveld refinement. The lattice distortion ($\varepsilon$) can be calculated by Equation (14) [68],

$$\varepsilon = \beta / 4\tan \theta \tag{14}$$

The crystallite size can be obtained from the widening of the XRD peaks by using Scherrer's formula as shown in Equation (15).

$$d = k\lambda / \beta\cos\theta \tag{15}$$

where $\beta$ is the full-width at half-maximum (FWHM) of the diffraction peak, k is the X-ray wavelength (nm), and $\theta$ is the diffraction angle [69]. The values of these variables are summarized in Table 3 [70].

**Table 3.** Unit cell values of XRD data of un-doped and doped $TiO_2$ [70].

| Samples | Phase | Crystallite Size | $\alpha$ (Å) | c (Å) | V (Å) | Lattice Distortion |
|---|---|---|---|---|---|---|
| $\alpha$-$Fe_2O_3$-$TiO_2$ | Anatase | 14 | 3.783 | 9.480 | 135.67 | 0.0111 |
| $\alpha$ -$Fe_2O_3$-$TiO_2$:Co | JCPDS | 12 | 3.785 | 9.506 | 136.18 | 0.0130 |
| $\alpha$ -$Fe_2O_3$-$TiO_2$:Cu | 01-089- | 11 | 3.787 | 9.508 | 136.36 | 0.0146 |
| $\alpha$ -$Fe_2O_3$-$TiO_2$:Bi | 4921 | 10 | 3.797 | 9.516 | 137.19 | 0.0150 |

UV/Vis diffuse reflectance spectrum (DRS) is applied to determine the band gap ($E_{bg}$) of the photocatalysts. The band gaps of these photocatalysts can be calculated with the Kubelka-Munk function as shown below,

$$(\alpha h\nu)^{2/n} = A(h\nu - E_g) \tag{16}$$

Previous literature reported that the measured absorption bands of $TiO_2$ and $Bi_2O_3$ are approximately 387.5 and 525.1 nm, corresponding to the band gaps of 3.20 ($TiO_2$) and 2.48 eV ($Bi_2O_3$), respectively (Figure 10a) [71]. After doping rGO to $TiO_2/Bi_2O_3$ photocatalyst, the light absorption edge of rGO-($Bi_2O_3$-$TiO_2$) redshifted to 429.06 nm, approaching the band gap of rGO-($Bi_2O_3$-$TiO_2$) to the visible light spectrum. The redshift is probably attributed to the 3-D hierarchical hetero-structures of $Bi_2O_3$ grain particles grown on the $TiO_2$ branches [71,72]. Multiple reflections and absorptions significantly improved the light absorption of rGO-($Bi_2O_3$-$TiO_2$). The interfacial partial charge transfer between $Bi_2O_3$ and $TiO_2$ catalysts caused a redshift phenomenon. This redshift process is thermodynamically feasible and dramatically enhances the photocatalytic activity in the wavelength region of visible light [73].

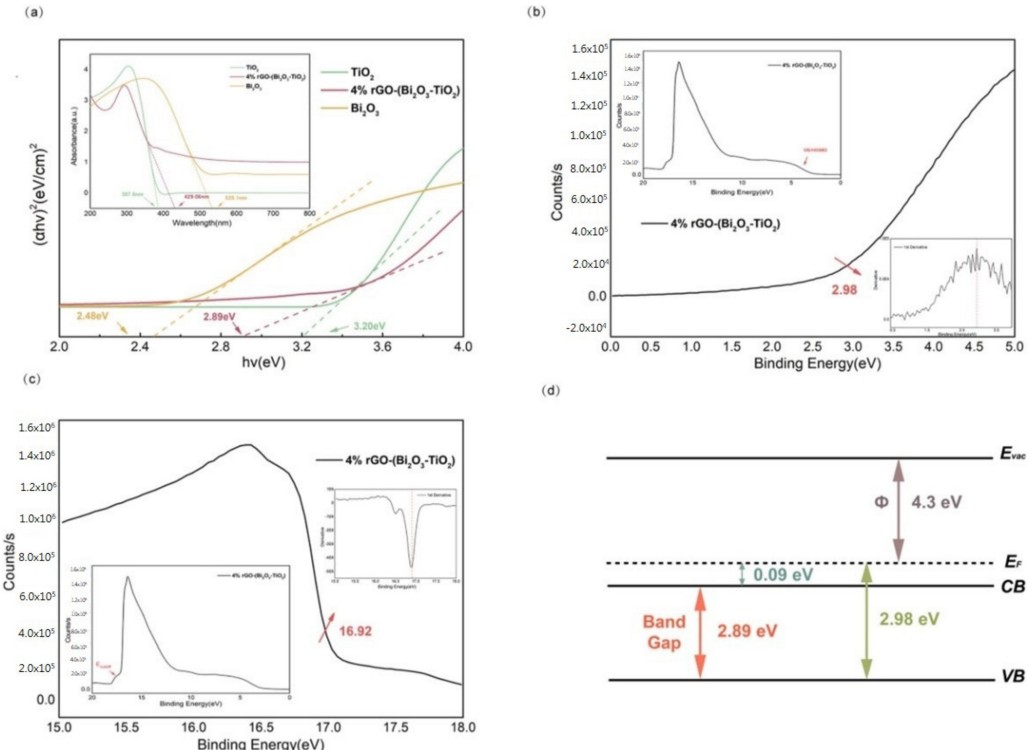

**Figure 10.** (**a**) UV/Vis DRS spectra of the as-prepared photocatalysts, (**b**,**c**) UPS spectrum of rGO-(Bi$_2$O$_3$-TiO$_2$), and (**d**) bandgap model of rGO-(Bi$_2$O$_3$-TiO$_2$) [71].

### 5.2. Physical Structure of Photo(electro)catalysts

In addition to the chemical properties of photo(electro)catalysts, their physical structure plays crucial roles on their photo(electro)catalytic activity. Particularly, nano- shape and preferential surface orientation of the facets play a key role in the electrochemical oxidation process, since the facets could affect not only the reactants themselves, but also the interfacial charge transfer. Inter-comparison with the larger counterparts, it is likely shown that the nano-sized photocatalysts suffer from an increased charge recombination rate on the catalyst surface, due to the large density of surface states acting as the carrier trapping sites. Oppositely, the larger-sized catalysts could sustain a built-in electrical potential within the catalytical particles, which is able to form a space charge region that facilitates charge separation, thus preventing the surface recombination of photo-induced $e_{cb}^-$/$h_{vb}^+$. Consequently, retarding the recombination of photo-induced charge carriers is essential for increasing the net charge transfer efficiency at the semiconductor electrolyte interface (SEI). Several parameters could influence the overall electrochemical response, including the electrode thickness, the specific surface area (SSA), and the grain size. Adequately controlling these operating parameters is crucial to enhance the adsorptive capacity [74].

Moreover, constructing heterojunctions by coupling $\alpha$-Fe$_2$O$_3$ with another semiconductor is an effective approach to improve the performance of photo(electro)catalytic water splitting [75]. The heterojunctions can facilitate charge separation by introducing an external electrical field at the interface between two semiconductors, thereby suppressing the recombination rate of photo-induced $e_{cb}^-$/$h_{vb}^+$ [75,76]. For instance, the $\alpha$-Fe$_2$O$_3$ doped TiO$_2$ heterojunctions formed using physicochemical vapor deposition processes have been investigated as photo-anodes and have exhibited improved photo(electro)catalytic activities for water splitting [37,77]. Even though $\alpha$-Fe$_2$O$_3$/TiO$_2$ heterojunction presents an energy barrier of about 0.6 eV for hole injection from $\alpha$-Fe$_2$O$_3$ into TiO$_2$ due to the misalignment of the valence band edges (VBEs), which creates a barrier at the interface of $\alpha$-Fe$_2$O$_3$/TiO$_2$ for interfacial tunneling of photo-induced $h_{vb}^+$, thus limiting the photo(electro)catalytic water oxidation performance.

Typically, the deposition of co-catalysts (e.g., metals or metallic oxides) on the photo (electro)catalysts demonstrates the enhanced PC and PEC activity once if an appropriate co-catalyst is employed (Figure 11a,b). The types of co-catalysts highly rely on their specific applications. For example, Pt-based co-catalysts play as an excellent electron sink for hydrogen evolution reaction [78,79], and Cu- or Ni-based co-catalysts are applied for $CO_2$ reduction reaction [80]. Due to the distinct signature of charge transfer in the PC and PEC reaction systems, the preparation of co-catalyst loaded photocatalysts could differ from co-catalyst deposited photo-electrodes. As depicted in Figure 11a,b, both oxidative and reductive co-catalysts could be loaded on the same photocatalysts (either n-type or p-type) applied in a suspension system. Moreover, these co-catalysts must be deposited separately on the n-type or p-type photocatalysts as the photo-anode for the oxidation process and the photo-cathode for the reduction process, respectively, in a PEC reaction system. Despite the difference of semiconductor types, the surface chemistry of the co-catalysts acting as active sites and its interaction with the photocatalyst further plays a crucial role in facilitating both PC and PEC systems [81].

In addition, other components can be doped/modified to play the roles as photoactive semiconductors (to produce plasmatic composites, Type I, Type II, or Z-scheme configuration), co-catalysts (to enhance surface kinetics or to serve as a passive layer), or metallic/non-metallic conductors (to accelerate electronic transfer). In brief, the multiple-component photoactive composites can offer four valuable photocatalytic benefits for the following aspects: (1) extending/shifting the light absorption spectrum to visible light (i.e., redshift phenomenon), (2) promoting charge separation of photo-induced $e_{cb}^-/h_{vb}^+$, (3) minimizing/retarding charge losses through the recombination of $e_{cb}^-/h_{vb}^+$, and (4) passivating the photo-corrosion of photoactive semiconductors.

As a nano-sized semiconductor photocatalyst that is coated with a conformal layer of an added component, a core-shell structure is constructively formed. The chemical composition of distinct surface introduced by the key component can also invoke the four aforementioned benefits. For example, encapsulating an unstable photoactive semiconductor (such as $Cu_2O$) within another semiconductor shell can enhance its physicochemical stability against photo-corrosion (i.e., photo-induced oxidative or reductive dissolution) [82]. In addition to having the photocatalyst as core structure, coating the photoactive semiconductor as an outer layer appears to be beneficial to the photocatalytic activity under certain conditions, as depicted in Figure 11c. By designing different configuration of the core and shell components, the equilibrium Fermi energy level can be tailored so that the photo-induced $e_{cb}^-/h_{vb}^+$ can be accumulated in the core, while charge transfer between the shell layer and the aqueous medium is promoted. The separation of photo-induced $e_{cb}^-/h_{vb}^+$ on the basis of band energy alignment prolongs charge lifetime, which would be potentially beneficial to the overall photocatalytic activity [79,83].

In order to establish a core-shell structured photo-electrode, the shell semiconductor usually suffers from a shorter minority charge transfer/diffusion length than the optical penetration path. Although the entire photo-electrode is capable to absorb the incident light with sufficient wavelength, the extremely short charge diffusion length is a crucial bottleneck for charge consumption and photo-activity. Therefore, charge extraction at the photo-generation sites on the surface of the primary shell semiconductor can be useful for transporting the photo-induced charges to a secondary component in the core semiconductor while they enable maximum light absorbance (MLA) [14,84]. In order to passivate the photo-dissolution of an unstable semiconductor, the designated core–shell structure may not be the solely strategy whereby the more commonly adopted and facile layer-by-layer coating approach may be sufficient (Figure 11d).

In addition to the widely established motivation in the formation of a heterojunction composite for the extended light absorption range (i.e., redshift phenomenon) and the prolonged separation of photo-induced $e_{cb}^-/h_{vb}^+$ (i.e., retarding effect), introducing a secondary component to a thin-film photo-electrode that plays a key role in passivating the surface of the underlying semiconductors. Unlike the core-shell photocatalysts

that are relatively hard to form shell layers with complete coverage, the layer-by-layer deposition of secondary components on the top of the photoactive semiconducting thin film can be achieved by using various techniques, including high-cost deposition techniques such as physical vapor deposition (PVD), chemical vapor deposition (CVD), atomic layer deposition (ALD), and low-cost techniques such as electrodeposition, photoelectron deposition, and solution-based coating methods. For example, a relatively inert metal oxide thin layer (e.g., $Ga_2O_3$ and $Al_2O_3$) has been applied to deposit on an n-type hematite photo-anode using chemical bath deposition [85]. Replacing solid-liquid interface (i.e., hematite-water interface) with a solid-solid interface has been a guiding principle commonly used to minimize the recombination rate of photo-induced $e_{cb}^-/h_{vb}^+$ or the chemical reactions over the surface of protected semiconductors. Briefly speaking, the mechanisms involving in the photo-stability improvement could beneficially achieve better photocatalysis, higher photo-voltage, and longer duration of $e_{cb}^-/h_{vb}^+$ recombination, with each mechanism interrelated.

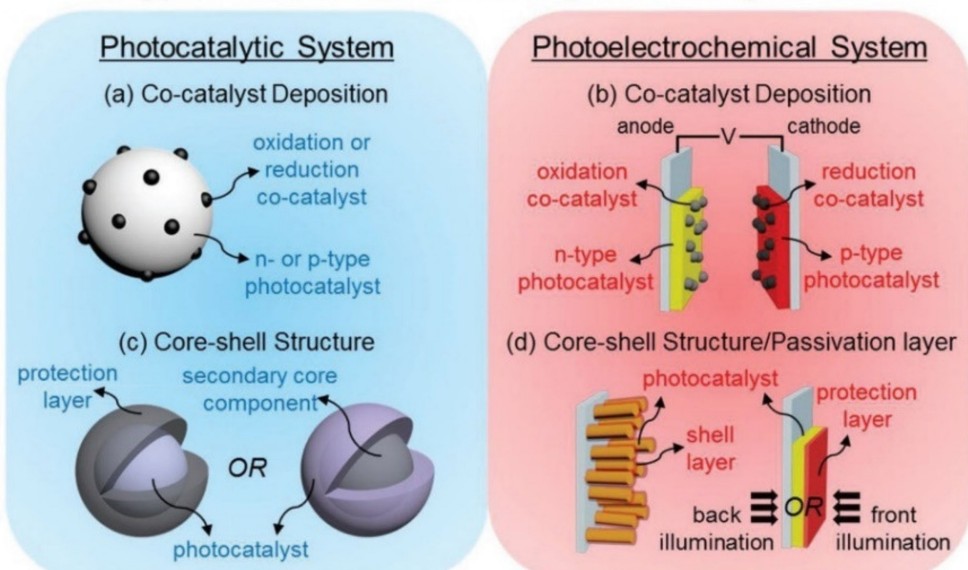

**Figure 11.** Typical structures of multiple component composites for the PC and PEC reaction systems. (**a,b**) co-catalyst deposition for powdered photocatalyst and thin-film photo-electrodes, (**c**) core–shell structure for powdered photocatalyst, and (**d**) core–shell structure and passivation layer of photo-electrodes [86].

Figure 12a–c depict the morphologies of ZnO, $BiVO_4$, and $ZnO/BiVO_4$ nanocomposites observed with a SEM and a TEM. It shows that ZnO nanorods grow vertically on the Fluorine-doped tin oxide (FTO) substrate, forming a typical hexagonal prism nano-sized array structure with an inside diameter of approximately 50 nm as illustrated in Figure 12a [87]. A plenty of voids are observed in the between of ZnO nanorods, which enlarges the specific surface area (SSA) and facilitates the introduction of $BiVO_4$ subsequently. Additionally, $BiVO_4$ directly deposited on the FTO substrate exhibits a regular short-bar structure with grain size of approximately 76 nm and creates a nanoporous reticular structure as shown in Figure 12b. As illustrated in Figure 12c, $BiVO_4$ deposited on the ZnO nanorods covers with coral-like $BiVO_4$ structure. Owing to the three dimensional (3-D) coral structure, $ZnO/BiVO_4$ nanocomposite enhances light absorption and accelerates the transfer of mass and electron, which thus evidently improve the PEC reactive performance [88].

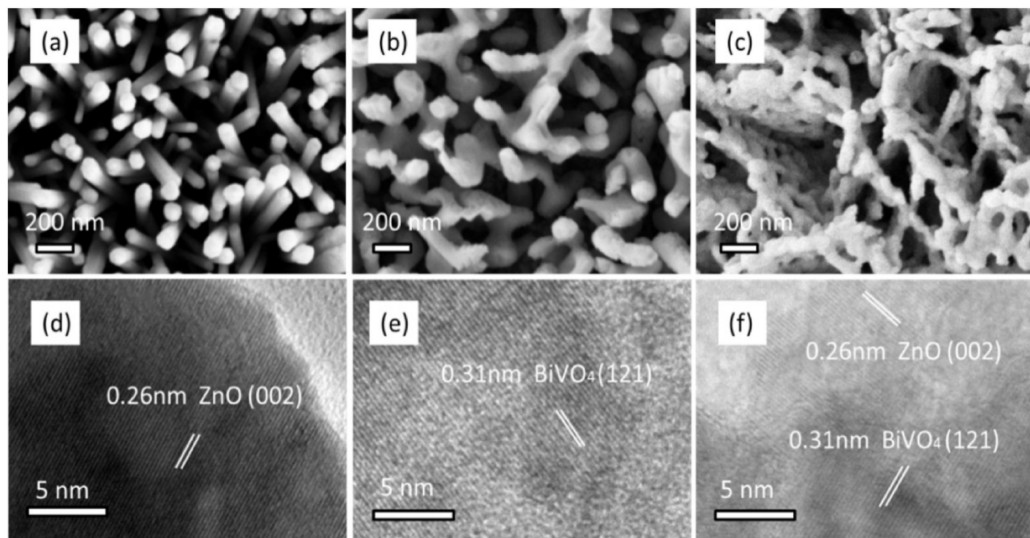

**Figure 12.** SEM and TEM images of (**a**,**d**) ZnO, (**b**,**e**) BiVO$_4$, and (**c**,**f**) ZnO/BiVO$_4$ nanocomposite [87].

Moreover, the TEM images of lattice fringe for ZnO, BiVO$_4$, and ZnO/BiVO$_4$ nanocomposite are depicted in Figure 12d–f. Figure 12d illustrates a lattice spacing of 0.26 nm is assigned to the crystal spacing of ZnO(002) facet in the hexagonal ZnO [88,89]. The lattice fringes of BiVO$_4$ in Figure 12e show that a crystal spacing of 0.31 nm is a typical characteristics of (121) facet in the monoclinic sheltie BiVO$_4$. Figure 12f displays the lattice fringes of the nanocomposite where the space of 0.26 and 0.31 nm are corresponding to the facets of the ZnO (002) and BiVO$_4$ (121), respectively. The nanocomposite forms an interface between the (002) facet of ZnO and the (121) facet of BiVO$_4$, which is conducive to the photogenic carrier migration between these two semiconductors. The intimate contact between ZnO and BiVO$_4$ in the nanocomposite would be beneficial to the effective transfer and separation of charges [37].

## 6. External Electrical Field Assisted Photocatalytic Techniques

### 6.1. Types of External Electric Field

In addition to the physicochemical properties of the photo(electro)catalysts and the types of photo(electro)catalytic technologies, the types of external electric fields exerted on the photo(electro)catalytic technologies play important roles on enhancing the degradation efficiency of multiphase pollutants. Several different external electric fields combining with photo(electro)catalytic oxidation technologies have been investigated in the past and are reviewed in the following [90–92].

Esbenshade et al. (2010) [90] once proposed an alternative method based on the principle of PEC oxidation using self-organized TiO$_2$ nano-tubular array electrodes irradiated by UV light to degrade sunscreen compounds (e.g., 4-methylbenzylidene camphor; 4-MBC) from the swimming pool water. Experimental results show that the PEC decomposition of 4-MBC is greatly improved with the application of electrical potential. When an electrical potential is applied, the competing process of photo-induced $e_{cb}^-/h_{vb}^+$ recombination is minimized/retarded by the resulting movement of the electrons to the counter photo-electrode. Thus, photo-induced $h_{vb}^+$ created by the photo-oxidation process would be trapped on the surface of the photo(electro)catalysts in order to create ·OH radical, a vital reactant to degrade 4-MBC by passing through the UV filters. This concludes that the improved photocatalytic activity in the PEC process results from the effective separation and retardation of $e_{cb}^-/h_{vb}^+$ by the applied positive bias, which forms an external electric field within the TiO$_2$ nano-tubular arrays and makes the photo-induced $e_{cb}^-/h_{vb}^+$ diffusing in the reverse directions [90].

Previous literature once reported the performance of PEC process on the oxidative degradation of acid green 50 textile dye (AG50) under the external electrical field applied

to the Ti/TiO$_2$-NT photo-anode [91]. The photo-anode is prepared with well-organized TiO$_2$ nanotubes (TiO2-NT) and electrochemically formed by anodizing titanium foil with the inside voltages ranging from 25 to 55 V in a non-aqueous electrolyte. The influential operating parameters including annealing temperatures of TiO$_2$ nanotubes (450, 600, and 800 °C) and external applied electrical potential (0.5 –1.5 V) at a constant pH value in the electrolyte are investigated for the decomposition of AG50 by TiO$_2$-NT PEC under the irradiation of UV light. Experimental results obtained from the degradation performance of PEC technique are further compared with that of PC technique. It shows that the PEC technique is superior to the PC technique in decomposing AG50 at the external applied electrical fields. It is mainly ascribed to the fact that ·OH radicals are the supreme oxidants, formed in the reactions (17)–(24), which are responsible for the degradation of AG50. Clearly, the application of external electrical potential could improve the performance of PC on the degradation of AG50 directly in the TiO$_2$ nanotubes [93]. The creation of an external electrical field allows the separation of photo-induced $e_{cb}^-/h_{vb}^+$ by transferring charges in different directions via the anode and the cathode and thereby retarding the recombination of photo-induced $e_{cb}^-/h_{vb}^+$ [91].

$$e_{cb}^- + h_{vb}^+ \rightarrow \text{heat} \tag{17}$$

$$O_{2\ ads} + e_{cb}^- \rightarrow O_2^- \tag{18}$$

$$O_2^- + H^+ \rightarrow \cdot HO_2 \tag{19}$$

$$\cdot HO_2 + e_{cb}^- \rightarrow HO_2^- \tag{20}$$

$$\cdot OH + e_{cb}^- \rightarrow OH^- \tag{21}$$

$$HO_2^- + H^+ \rightarrow H_2O_2 \tag{22}$$

$$H_2O + h_{vb}^+ \rightarrow \cdot OH + H^+ \tag{23}$$

$$OH^- + h_{vb}^+ \rightarrow O \cdot H \tag{24}$$

Mais et al. (2019) investigated the photoelectrocatalytic oxidation of phenol in the aqueous solution under the irradiation of solar light with polyaniline (PANI)-modified TiO$_2$ electrodes. In their study, an innovative TiO$_2$/PANI electrode is proposed for the PEC oxidation of organic pollutants, which is obtained through a layer-by-layer electrochemical approach. Previous study [91] reported that phenol can be converted to aromatic, then decomposed to aliphatic, and finally oxidized to CO$_2$. The oxidation processes are illustrated in Figure 13 [94]. A surface grafting by the reduction of 4-nitrobenzendiazonium salt is followed by the reduction of nitro-groups to amino-groups. PANI is electrodeposited on the under-layer. The results show that the photo-induced $h_{vb}^+$ may not only oxidize water molecules (H$_2$O) to produce ·OH radicals, but might also directly oxidize organic pollutants. Since heterogeneous reactions are chemically involved, a certain coverage degree of the reactants is required at the surface of photo-electrode. Based on the experimental results, the highest oxidation efficiencies of phenol are achieved at the low initial phenol concentration. Therefore, the phenol coverage degree is low and a high concentration of active sites is available on the surface of photo-electrode. Water oxidation may occur as the main process to produce ·OH radicals if the photo-anode potential is sufficiently high, which activates the photo-oxidation of organic reactants, possibly adsorbed at the adjacent surface sites. When the phenol concentration is too high, the high coverage degree of phenol may limit the availability of active sites for water oxidation, and thus decompose phenol via a slow direct oxidation [92].

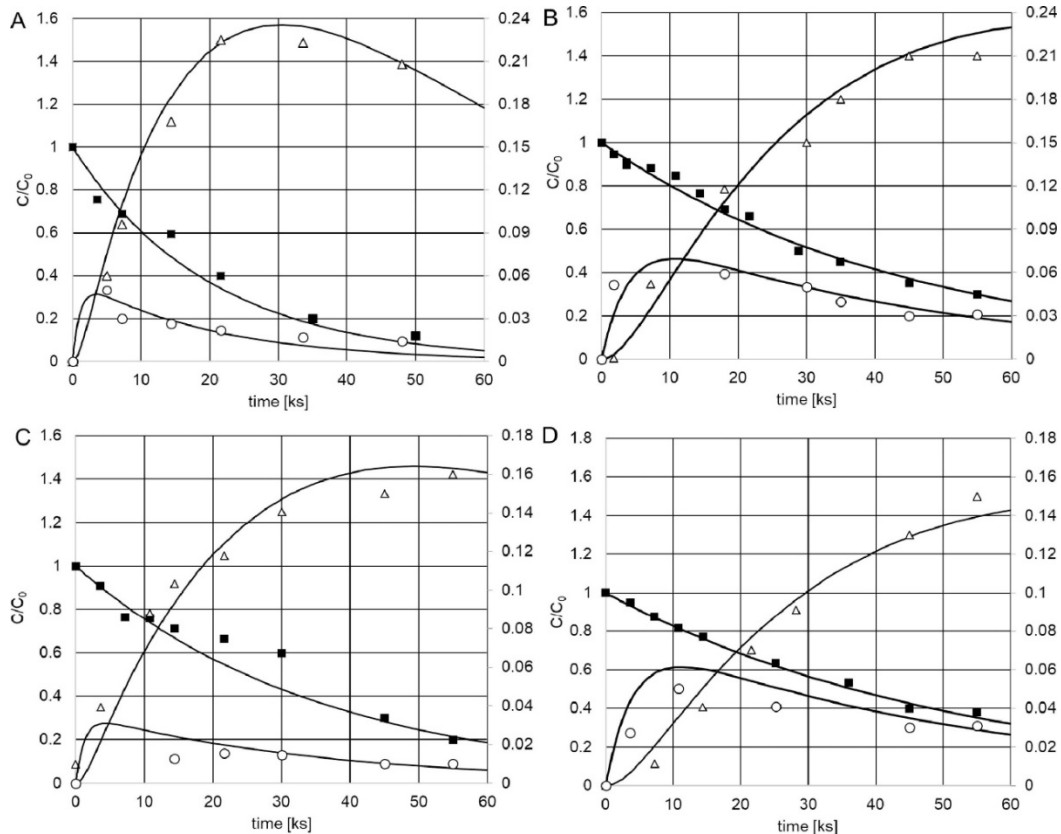

**Figure 13.** Comparison between experimental (symbols) and model predicted data (lines) of concentrations of phenol (full symbols), aromatic intermediates (circles, secondary axis) and aliphatic intermediates (triangles, primary axis). All the values are normalized with the initial phenol concentration. Data obtained with $C_0$ = 300 mg/cm$^3$ under different experimental conditions: i = 0.18 mA/cm$^2$, NT-PANI (**A**); i = 0.18 mA/cm$^2$, NT (**B**); i = 0.09 mA/cm$^2$, NT-PANI (**C**); i = 0.09 mA/cm$^2$, NT (**D**) [89].

### 6.2. Efficiency of External Electrical Field

Ju et al. (2021) [95] once investigated the reaction mechanisms for the decomposition of Bisphenol A by PEC technique using new semiconductors as the working photo-electrodes. As illustrated in Figure 14a, C60@AgCl-LDO nano-sized photocatalysts with an applied electrical voltage of −1.0 eV show higher efficiency in Bisphenol A (BPA) reduction than that with an applied electrical voltage of +1.0 eV by irradiating with visible light (300 W Xenon lamp). It indicates that C60@AgCl-LDO functions more efficiently as a photo-anode than as a photo-cathode for the reduction of BPA. The semiconductor as a photo-anode prolongs the separation duration of photo-induced $e_{cb}^-/h_{vb}^+$, which allows the formation of a large amount of ·OH radicals.

As depicted in Figure 14b, 8% of BPA is reduced by photocatalysis (PC), while 75% of BPA is reduced by electrocatalysis (EC), after the PEC reaction duration of 120 min. Using C60@AgCl-LDO nano-sized photocatalysts with an applied electrical voltage of −1.0 eV, 99% of BPA can be reduced by the irradiation of visible light after the PEC reaction duration of 120 min due to a synergic effect of PC and EC combinations. It is mainly attributed to the fact that C60@AgCl-LDO/Pt photo-anode has a remarkable high electrochemical activity towards oxygen evolution, and thereby exhibits a high chemical reactivity for BPA oxidation. The C60@AgCl-LDO nano-sized photocatalysts show remarkably high efficiency of BPA in the heterogeneous PEC reaction system under the irradiation of visible light with an applied voltage of electrical field for BPA reduction than ZnAlTi-LDO nano-sized photocatalysts (see Figure 14c).

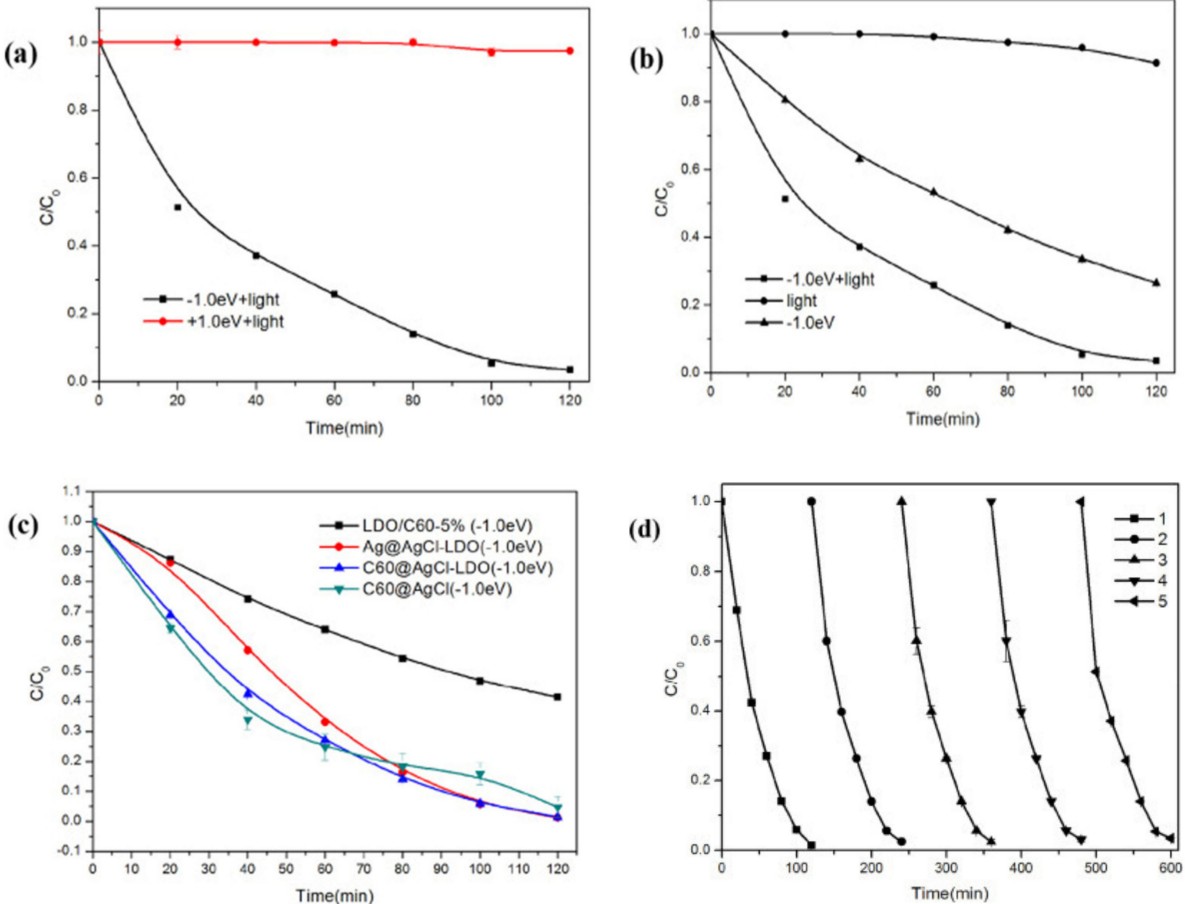

**Figure 14.** (**a**) The effect of positive or negative voltage on photoelectrocatalytic degradation of BPA. (**b**) The photoelectrocatalytic degradation of BPA by C60@AgCl-LDO photocatalysts. (**c**) The photoelectrocatalytic degradation of BPA by different materials (**d**) The photoelectrocatalytic degradation of BPA in consecutive runs using the recycled C60@AgCl-LDO photocatalysts. (Experimental conditions: 10 mg/L BPA, reaction dose 0.25 g/L, pH = 6.5, 0.1 mol/L NaCl, −1.0 eV, temperature 28 °C) [95].

As illustrated in Figure 14d, the degradation efficiency of BPA goes up to 95% by using C60@AgCl-LDO nano-sized photocatalysts in an environment of simulated lights with an applied electrical voltage of −1.0 eV for the PEC reaction duration of 120 min. Furthermore, C60@AgCl-LDO nano-sized photocatalysts are convenient for recycling and maintaining an excellent photoelectrical activity after five cycles. Additionally, Ag@AgCl-LDO and C60@AgCl nano-sized photocatalysts are automatically decomposed after one test and cannot be recycled [95].

Applied electrical potential is of great importance in photo(electro)chemical reactions since it could directly affect the processes of both electrical reaction and photo(electro)catalytic degradation efficiency. In order to investigate the influence of applied electrical potential on the PEC degradation of BPA, different electrical potentials are applied using C60@AgCl-LDO photocatalysts as the photo-anode in 0.1 mol/L of NaCl electrolyte at a pH of 6.5. As illustrated in Figure 15, there is a positive correlation between BPA degradation and applied electrical potential, particularly from 0 to 1.0 eV, due to a higher electrical potential which increases the amount of active species and accelerates the transfer of electrons, leading to enhance the photo-oxidization efficiency of BPA. As the applied electrical potential is further increased to above 1.0 eV, the photo-degradation of BPA would be reduced. However, when the applied electrical potential is too high, there are probably too many active species that might hinder the separation of photo-induced $e_{cb}^{-}/h_{vb}^{+}$, thereby decreasing the degradation efficiency of PEC [92].

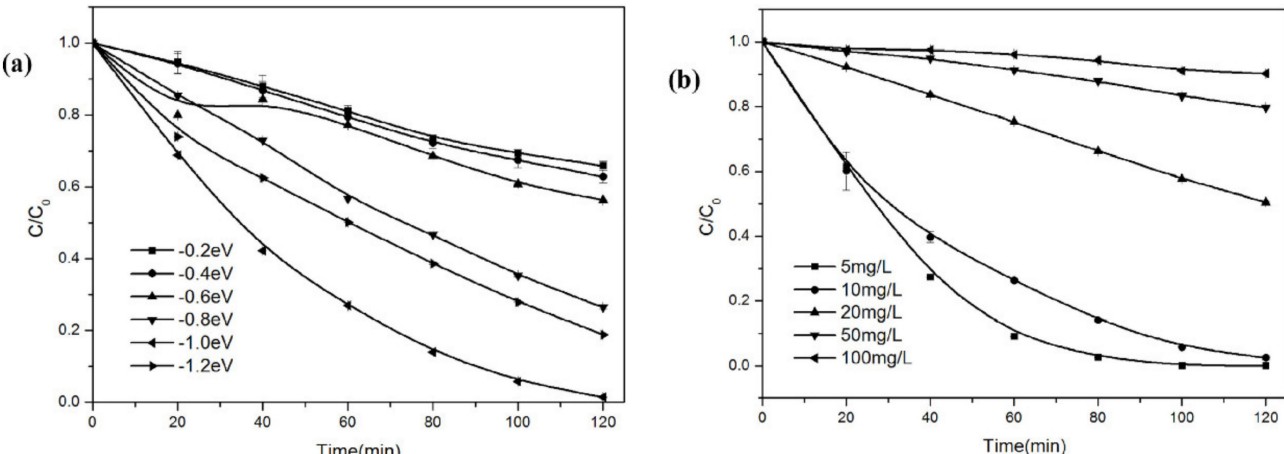

**Figure 15.** (**a**) The effect of different voltages on the photoelectrocatalytic degradation of BPA. (**b**) The effect of different pollutant concentration on the photoelectrocatalytic degradation of BPA by C60@AgCl-LDO photocatalysts (Experimental conditions: 10 mg/L BPA, reaction dose 0.25 g/L, pH = 6.5, 0.1 mol/L NaCl, −1.0 eV, temperature 28 °C) [95].

According to the research of Ju et al., 2021, it shows photoelectrocatalytic could enhance the efficiency of pollutants degradation. Table 4 shows the photoelectrocatalytic activity of different materials. In the following we will introduce the performance of them.

Ensaldo-Rentería et al. (2018) [91] further investigated the performance of PEC technique on the photo-oxidation of AG50 under the irradiation of UV light at different bias potential (0.5, 1.0, 1.2, and 1.5 V). In addition, secondary reactions such as photo-cathodic hydrogen evolution and photo-cathodic reduction of $H_2O_2$ to $H_2O$ may affect the photo-degradation of AG50 at the electrical potentials above 1.2 V. Nevertheless, the application of 1.0 and 1.2 V of cell potential yields the degradation of 76 and 96% of AG50 at the photo-oxidation reaction of 150 and 360 min, respectively. It might be attributed to $H_2O_2$ production at the applied electrical potential since oxygen atoms can be reduced on carbon graphite.

Liu et al. (2020) [96] reported that a well-designed hierarchical F-BiVO4@NiFe-LDH photo-anode with a core-shell hetero-structure has been successfully fabricated by F⁻ ions etching and in-situ electrochemical synthetic process of NiFe-LDH onto $BiVO_4$ film. Both theoretical and experimental results indicate that a synergistic effect of F⁻ doping on NiFe-LDH could simultaneously improve the light absorption, charge separation, and charge injection efficiency of F-BiVO₄@NiFe-LDH photo-anode. Accordingly, the maximum photocurrent density (MPD) of the fabricated F-BiVO₄@NiFe-LDH photo-anode at 1.23 V vs. RHE is about 6-fold higher than that of the pristine $BiVO_4$ photo-electrode. Moreover, the composite photo-anode was effectively applied to the PEC degradation of tetracycline hydrochloride, with the photo-degradation rate reaching 86% within 2 h and no further attenuation after 4 cycles. Evidently, it presents a sunlight-driven, efficient, and sustainable method for the photo-degradation of organic contaminants coupled with the formation of $H_2$. It is expected that the promising strategy would be extended to the fabrication of other novel photo-electrodes for advanced performance in the fields of solar conversion and wastewater treatment.

Table 4. Comparison of photoanodes from recent studies with their synthesis techniques, advantages, and efficiencies.

| Synthesis Techniques | Electrode Types | Advantages | Efficiencies | References |
|---|---|---|---|---|
| Electrochemical anodisation | Ti/TiO$_2$-NT | The photocatalytic reactivity of nanotube is influenced via applied electrical potential and annealing temperature at a fixed pH of solution. | The use of 1.0 V and 1.2 V of cell potential yielded 96% of acid green 50 (AG50) degradation at 360 min | [91] |
| Electrochemical synthesis process | F-BiVO$_4$@NiFe-LDH | The degradation rate of core-shell photo-anode is about 6 fold higher than pristine BiVO$_4$ photo-electrode. | The composite photo-anode was effectively applied to the PEC degradation of tetracycline hydrochloride, with the degradation rate reaching 86 %. | [96] |
| Solvothermal combined with hydrothermal | TiO$_2$/Bi$_2$WO$_6$/Au NRAs | The fabricated hierarchical TiO$_2$/Bi$_2$WO$_6$/Au NRAs achieve an enhanced photocurrent density of 29 μA/cm$^2$, about 11 times as high as that of pristine TiO$_2$ NRAs and 2.6 times larger than that of hierarchical TiO$_2$/Bi$_2$WO$_6$ NRAs. | The hierarchically heterogeneous TiO$_2$/Bi$_2$WO$_6$/Au architectures show remarkably enhanced photocatalytic degradation for methyl blue molecules and obtain photo-degradation of 91.5%, much higher than that (50.3%) of pure TiO$_2$ NRAs. | [67] |
| Pulsed laser deposition | α-Fe$_2$O$_3$/Au/TiO$_2$ | The photocurrent density is increased by about 4-folds for photoelectrocatalytic water splitting. Photo-anode yields a maximum photocurrent density of 1.05 mA/cm$^2$ | The Faradaic efficiencies for H$_2$ and O$_2$ evolution are 93.5 and 91.6% | [97] |
| Electro-deposition | ZnO/BiVO$_4$ | Due to its coral structure, the nanocomposite increases the light absorption and mass transfer of visible light | The photoelectrocatalytic degradation efficiency of tetracycline was up to 84.5%. | [87] |

Yao et al. (2019) [67] reported that the hierarchical $TiO_2/Bi_2WO_6/Au$ NRAs are fabricated through a simple solvothermal route combined with hydrothermal method. The light absorption indicates that the hierarchical $TiO_2/Bi_2WO_6/Au$ NRAs show the improvement of visible light absorption. Spectroscopic tests provide direct evidences for the formation of a type II band alignment. Benefiting from the synergistic effects of the type II band alignment and SPR effect, the fabricated hierarchical $TiO_2/Bi_2WO_6/Au$ NRAs achieve an enhanced photocurrent density of 29 $\mu A/cm^2$, which is approximately 11 times as high as that of pristine $TiO_2$ NRAs and 2.6 times larger than that of hierarchical $TiO_2/Bi_2WO_6$ NRAs at 1 V vs Ag/AgCl. The hierarchical $TiO_2/Bi_2WO_6/Au$ NRAs reveal an excellent stability under a 2 h light irradiation. Additionally, the hierarchically heterogeneous $TiO_2/Bi_2WO_6/Au$ architectures show remarkably enhanced photocatalytic degradation for membrane blue (MB) molecules and obtain photodegradation of about 91.5%, which is much higher than that (50.3%) of pure $TiO_2$ NRAs. The work achieved in the study offers a promising strategy for constructing visible-light photocatalyst by forming the type II band alignment and integrating surface plasma resonance (SPR) effect.

Fu et al. (2020) [97] reported that the photo-stability of photo-anodes is assessed by measuring the photocurrent density at a constant electrical potential of 1.23 $V_{RHE}$ under the simulated solar irradiation over 12 h (see Figure 16a). All of the photo-anodes display excellent photo-stability, with photocurrent density remaining at nearly 100% of the initial values over a 12 h experimental test. The quantities of $O_2$ and $H_2$ produced by $\alpha$-$Fe_2O_3/Au/TiO_2$ photo-anode and counter photo-electrode are measured by using gas chromatography (GC) during the 12 h electrolysis under the experimental conditions identical to those of the photo-stability test. As depicted in Figure 16b, gases are evolved at a constant photocatalytic reaction rate over the course of the experimental tests. The reaction rates of $H_2$ and $O_2$ evolution are calculated as 18.67 and 9.24 $\mu mol/cm^2 \cdot h$, respectively, yielding the stoichiometric ratio of 2:1 expected for water splitting. The Faradaic efficiencies of $H_2$ and $O_2$ evolution are 93.5 and 91.6%, respectively, and the slightly smaller Faradaic efficiency of $O_2$ evolution is due to the relatively greater dissolution of $O_2$ in the electrolyte. Moreover, there are no obvious difference in the morphology and chemical composition in $\alpha$-$Fe_2O_3/Au/TiO_2$ after the PEC measures, further confirming the excellent chemical stabilization of photo-anode. In combination, all of the results demonstrate that $\alpha$-$Fe_2O_3/Au/TiO_2$ photo-anode is highly active and much stable for the PEC technique of water splitting.

Li et al. (2021) [87] reported that coral-like ZnO nanorod/$BiVO_4$ nanocomposite are successfully fabricated on the FTO substrate via chemical bath deposition and electrodeposition. The nanocomposite yields a photocurrent of about 0.29 $mA/cm^2$ at 0 V vs. Ag/AgCl under the irradiation of visible light ($\lambda \geqq 420$ nm). The degradation efficiency of tetracycline from the ZnO/$BiVO_4$ nanocomposite is 84.5% within 60 min of PEC technique after 30-min adsorption in the dark condition. Due to its coral structure, the nanocomposite increases the absorption and mass transfer of visible light and effectively inhibits the photogenic carrier recombination, thus improving the photocatalytic activity. Trapping and electron paramagnetic resonance (EPR) measures suggest that the PEC activity of ZnO/$BiVO_4$ mainly originated from the pronounced variation of capability produces the radicals of $\cdot O_2^-$ and $\cdot OH$ upon visible light irradiation. The cyclic experiments indicate excellent stability and reusability of the nanocomposite. The ZnO/$BiVO_4$ nano-electrode is proven as a promising candidate for the PEC degradation of organic contaminants in water.

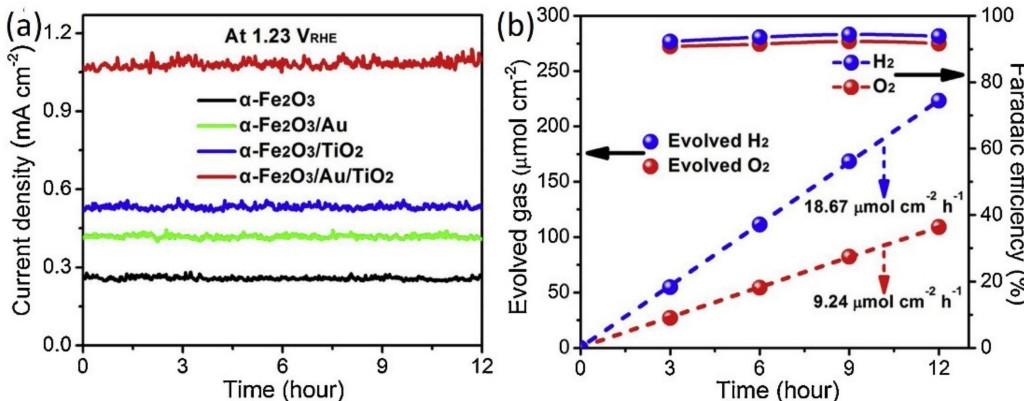

**Figure 16.** (**a**) Chronoamperometry for irradiated samples over a period of 12 h at 1.23 $V_{RHE}$. (**b**) Amounts of evolved gases and the calculated Faradic efficiencies using $\alpha$-Fe$_2$O$_3$/Au/TiO$_2$ photo-anode during the photo-stability test. All tests are measured under the front-side illumination (from the photo-anode side) [97].

## 7. Potential Application of Photo(electro)catalysis

Overall, photo(electro)catalytic technologies including photocatalytic oxidation (PCO) and photoelectrocatalytic oxidation (PECO) processes have been technologies potentially applied for the treatment of multi-phase pollutants in the wastewaters and waste gases. Photo(electro)catalytic oxidation process is a promising technique by using semiconductors as photocatalysts to convert light and/or electrical energy to chemical energy. Such novel PC and PEC techniques have been potentially applied for the abatement of organic and inorganic pollutants theoretically and practically.

### 7.1. Organic Compounds (OCs)

Organic compounds (OCs) in the wastewaters and waste gases can be effectively degraded by innovative AOTs such as PC and PEC techniques with applied electrical potentials. Moreover, organic compounds in the landfill leachates and volatile organic compounds (VOCs) in the waste gases have also been investigated in the past.

Landfill leachates possess high concentrations and huge amounts of organic contaminants, inorganic acids, heavy metals, and ammonia nitrogen as well [98]. Tauchert et al., (2006) [99] treats the aged landfill leachates with highly recalcitrant nature by using a photo(electro)chemical reaction system. His study further compares the usages of photolysis, heterogeneous photolysis, electrolysis, and photoelectrolysis (PE) in a photo(electro)chemical (PEC) reactor with dimensional stable anode (DSA) and the irradiation of 125-W UV mercury lamp. The degradation efficiency is achieved by the indirect electrochemical oxidation aided with oxidants such as ·OH, H$_2$O$_2$, and chlorine species, which are further photochemically transformed into more active radicals. It reveals that dark coloration of landfill leachate is the restraining factor in photoelectron-chemical process, which urges the requirement of some coagulative techniques before electrochemical advanced oxidation processes (EAOPs) for achieving recommendable removal efficiency. The schematic diagram of the UV-assisted PEC-MFC system is shown as Figure 17.

A novel UV-assisted PEC-MFC system has been applied for the treatment of ethyl acetate (EA) and toluene in the aqueous phase [34]. In this system, a CeO$_2$/TiO$_2$/ACF photocatalytic cathode with excellent photoelectrocatalytic properties and redox abilities is successfully prepared and integrated with bio-anode. The air purification system exhibits excellent elimination capacities (EC) for EA (0.39 g/m$^3$, EC: 2.52 g/m$^3$·h) or toluene (0.29–4.10 g/m$^3$, EC: 1.89–28.04 g/m$^3$·h) under the irradiation of UV light at room temperature. Moreover, UV-assisted PEC-MFC reaction system can continuously decompose OCs and simultaneously generate electricity. The possible mechanism of OC removal in the novel UV-assisted PEC-MFC reaction system is proposed and investigated. This study not only develops a CeO$_2$/TiO$_2$/ACF photo-cathode, but also provides an innovative AOT

system for the potential application on the effective treatment of OCs in the wastewaters in practice [34].

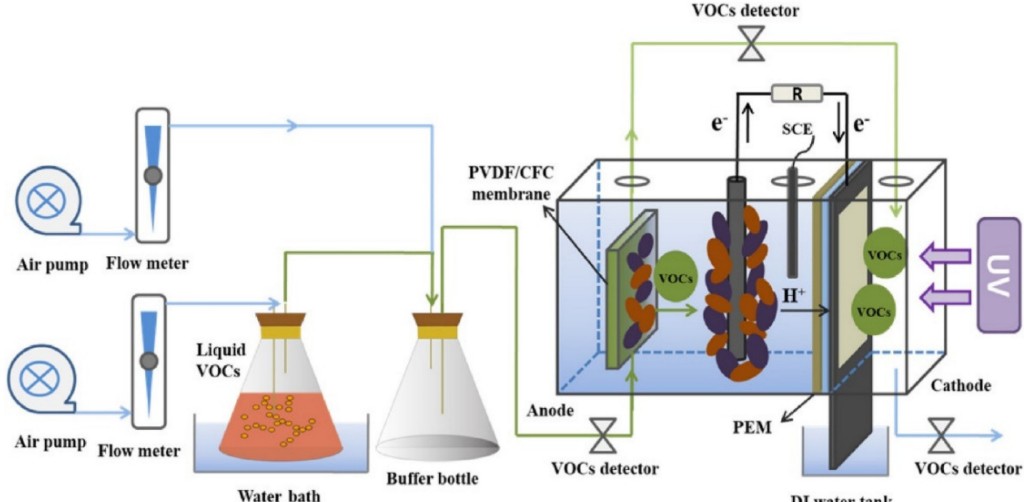

**Figure 17.** Schematic diagram of the UV-assisted PEC-MFC system [34].

An electrical glassfiber filters (EGFs) coated with nano-sized $TiO_2$ (i.e., $TiO_2$/EGFs) is developed to degrade VOCs in the waste gases by using the PEC technique (Li et al., 2017). Acetone is used as a typical organic compound for employing the VOC degradation experiments in a pan-type batch PEC reactor. The schematic diagram of the PEC reaction system is illustrated in Figure 18. The $TiO_2$/EGF-based PEC technique exhibits higher decomposition efficiency of acetone than that of traditional $TiO_2$ photocatalytic (PC) technique. The decomposition rate of acetone with the PEC technique is approximately 2.79 times faster than that with the PC technique. Moreover, the decomposition rate of acetone using the PEC technique increases with the applied external electrical field intensities from 0 to 6500 V. The correlation between the reaction rate of acetone and its initial concentration follows the pseudo first-order kinetics for acetone concentrations <100 ppm, but followed the pseudo zero-order kinetics for acetone concentrations >100 ppm [100].

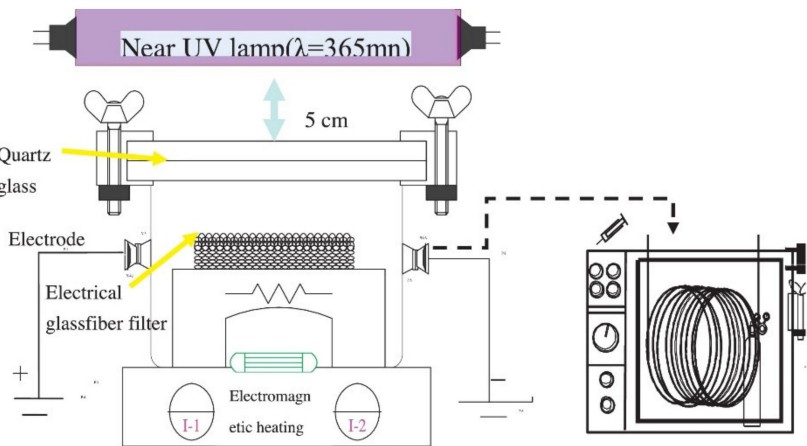

(1) Optical catalytic reaction system   (2) reactant and product analysis system

**Figure 18.** Schematic diagram of the PEC reaction system [100].

## 7.2. Metallic Compounds

In addition to the decomposition of organic compounds, metallic compounds can be oxidized by the innovative PC and PEC techniques. Zhang et al. (2020) [101] once propose a mechanism of a three-compartment cell system as shown in Figure 19. In the photo-anode cell, $PO_2^-$ is oxidized to $PO_4^{3-}$ via ·OH radicals produced by the TNA/NSS

photo-anode. In the photo-cathode cell, $Ni^{2+}$ reacts with $\cdot OH$ to form $\alpha$-$Ni(OH)_2$, which is then further reduced to metallic element Ni. Furthermore, the build-up of charge carriers in the photo-anode and photo-cathode could induce the transport of $H^+$ and $Cl^-$ from the photo-anode and photo-cathode cell to the middle cell and produces hydrogen chloride (HCl). Thus, this establishes a three-compartment PEC cell system to achieve $H_2PO_2^-$ oxidation and simultaneous metallic Ni recovery along with HCl production, which plays a key role in wastewater resourcezation treatment (WRT) process for the recovery of P and Ni from the electroless nickel plating (ENP) effluents [101]

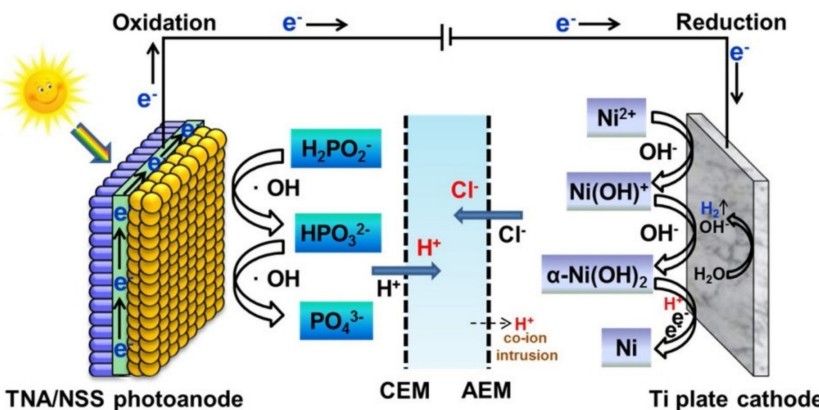

**Figure 19.** Proposed mechanism of the three-compartment cell system for $H_2PO_2^-$ oxidation, Ni recovery, and HCl production [101].

The treatment of actual electroless nickel plating (ENP) effluents by WRT process is investigated to check its applicability in terms of decomposition efficiency, cost-benefit, and stability assessment [101]. The effluents are initially pretreated by physical separation measures (PSMs) to obviously remove suspended solids in the wastewaters. The initial effluents mainly contain phosphorous (P) compounds ($H_2PO_2^-$, $HPO_3^{2-}$, and $PO_4^{3-}$) and nickel compound ($Ni^{2+}$). The WRT process is adopted to treat the effluents to achieve the recovery of P and Ni. The effluents are initially treated with ion exchange (IE). Thus, wastewater containing P compounds and effluents containing $NiCl_2$ are obtained, respectively. The treatment of P and Ni compounds by using PEC technique is conducted to oxidize both $H_2PO_2^-$ and $HPO_3^{2-}$ to $PO_4^{3-}$, and recovers Ni along with HCl production (see Figure 20). Finally, a chemical precipitation process (CPP) is further applied to recover ferric phosphate ($FePO_4$) from the effluents.

Feng et al. (2017) [102] once reported that the conversion of $Cr^{6+}$ over the TiHAP film by the PEC technique can achieve up to 84% under the irradiation of UV light for 90 min. As illustrated in Figure 21, the conversion efficiency of $Cr^{6+}$ over the TiHAP film are always higher than that over the $TiO_2$ film. The F-doped TiHAP film (TiFHAP) shows the enhanced PC and PEC activities than the TiHAP film, with the highest conversion of $Cr^{6+}$ for the optimal 5.6 wt% F-doped TiHAP film. A two-compartment system is employed to investigate the reaction mechanisms of the PEC process. Different from the traditional knowing that $Cr^{6+}$ would be reduced on photo-cathode by photo-induced $e_{cb}^-$ transferred from photo-anode photocatalyst. It suggests that $Cr^{6+}$ is mainly reduced on the surface of TiHAP photo-anode [102].

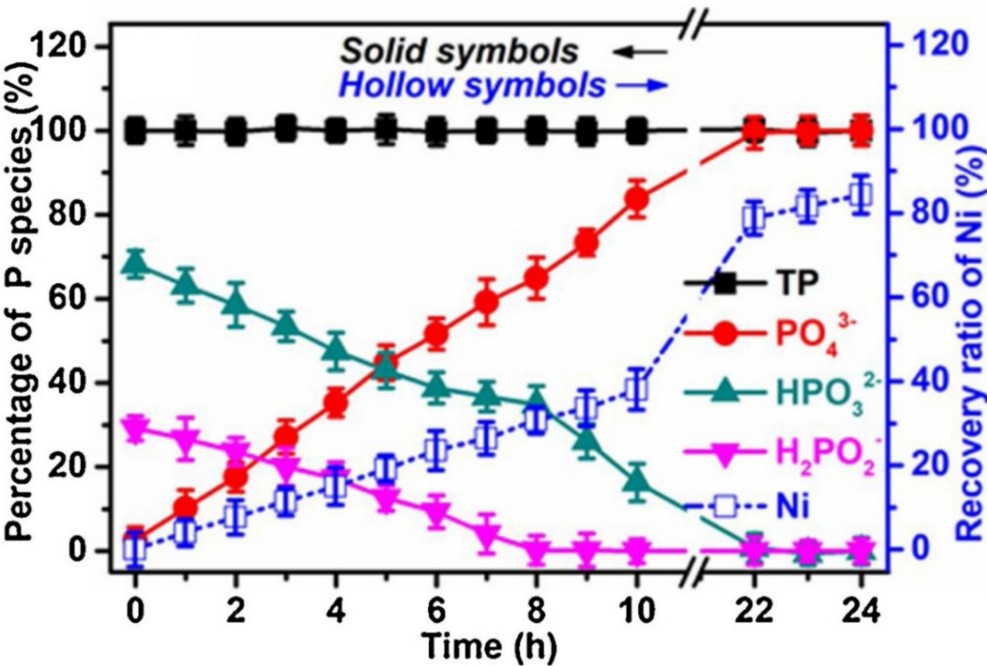

**Figure 20.** The performance of PEC treatment technique for actual ENP effluents [101].

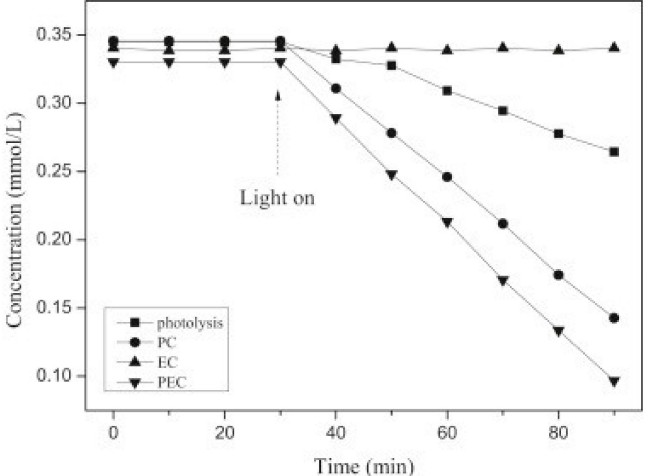

**Figure 21.** The variation of $Cr^{6+}$ concentration with reaction time over TiHAP film during the photolysis, PC, EC and PEC reactions [102].

Shen et al. (2016) [10] once conducted experimental tests to enhance the photo-oxidation efficiency of elemental mercury ($Hg^0$) by nano-sized immobilized $WO_3/TiO_2$ photocatalysts at high temperatures using a batch PC reaction system. The schematic diagram of photo-oxidation of $Hg^0$ by using $WO_3/TiO_2$ photocatalyst under the irradiation of UV light is illustrated in Figure 22. The photo-oxidation efficiencies of $Hg^0$ at 160 °C by $TiO_2$ (Degussa P-25), $TiO_2$ (sol-gel), and 17% $WO_3/TiO_2$ photocatalysts, respectively with an influent $Hg^0$ concentration of 25 μg/m$^3$ are explored. Experimental results clearly show that the photo-oxidation efficiency of $Hg^0$ is highly enhanced by $WO_3$ dopant at a high temperature of 160 °C, increasing from 14% for Degussa P-25 to 63% for $WO_3/TiO_2$. Additionally, different $WO_3$ doping amount affects the photo-oxidation efficiency of $Hg^0$, and the photocatalytic reactivity of 1–3% $WO_3/TiO_2$ is superior to 5–7% $WO_3/TiO_2$. The equilibrium constant ($K_{Hg0}$) and Gibbs free energy ($\Delta G$) of $Hg^0$ by photocatalytic reaction system using $WO_3/TiO_2$ are then determined by the L-H kinetic model [10].

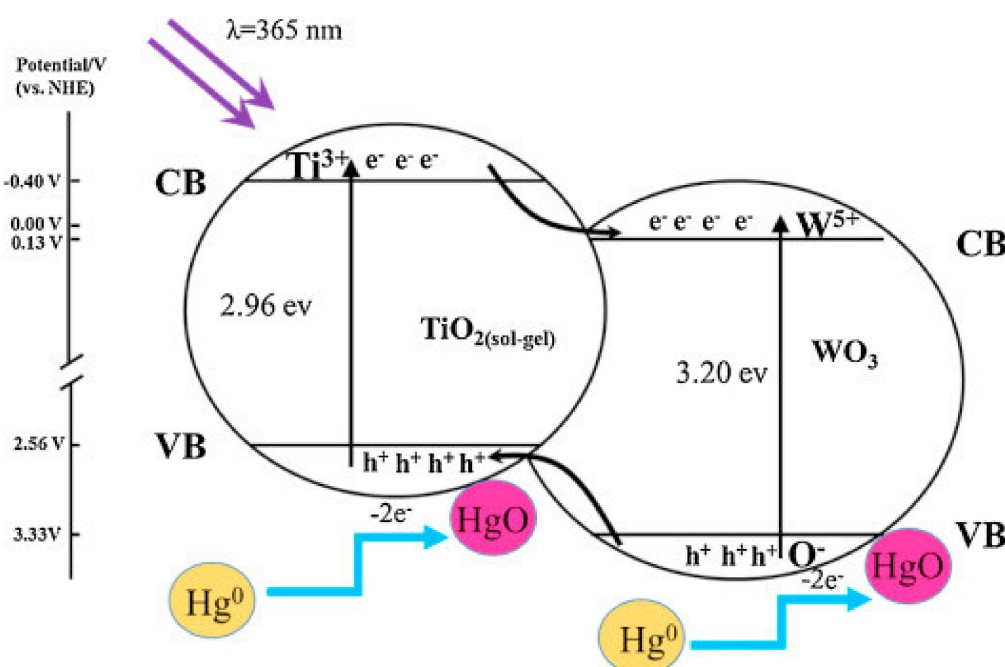

**Figure 22.** The schematic diagram of photo-oxidation of $Hg^0$ by using $WO_3/TiO_2$ under the irradiation of UV light [10].

### 7.3. Pharmaceutical Compounds

The degradation of pharmaceutical compounds has been investigated by various treatment processes such as adsorption, electrosorption, and photo(electro)catalysis in the past. Sheydaei et al. (2021) [103] once conducted a sorption (dark) experiment for the removal of cefixime with the g-$C_3N_4$/Ce-ZnO/Ti nanocomposite. The degradation efficiency of pharmaceutical compounds is approximately 7% for 180 min, indicating that the sorption of celfixine is negligible. With the application of external electrical potential on the g-$C_3N_4$/Ce-ZnO/Ti nanocomposite (i.e., electro-sorption), the degradation efficiency of cefixime increases slightly up to 16% after 180 min. Although the enhancement of is not high, but in comparison with the sorption, it might be attributed to the electrostatic interaction and affinity between celfixine on the surface of positively charged nano-composite and negatively charged cefixime molecules. Furthermore, as illustrated in Figure 23, the photo-degradation efficiency of cefixime using the photocatalytic process with the irradiation of visible light is relatively higher than both sorption and electro-sorption processes, which proves the efficient photocatalytic activity of the g-$C_3N_4$/Ce-ZnO/Ti nanocomposite. Further considering the experimental results, the degradation efficiency of cefixime up to 80% is obtained for the photoelectrocatalytic process under the irradiation of visible light. In fact, the synergistic effect between electro-sorption and photocatalytic activity leads to quick sorption of gaseous pollutants on the surface of photocatalysts to photocatalytic reaction with the photo-induced $h_{vb}^+$ to form surface ·OH, thus resulting in higher removal efficiency of cefixime [103].

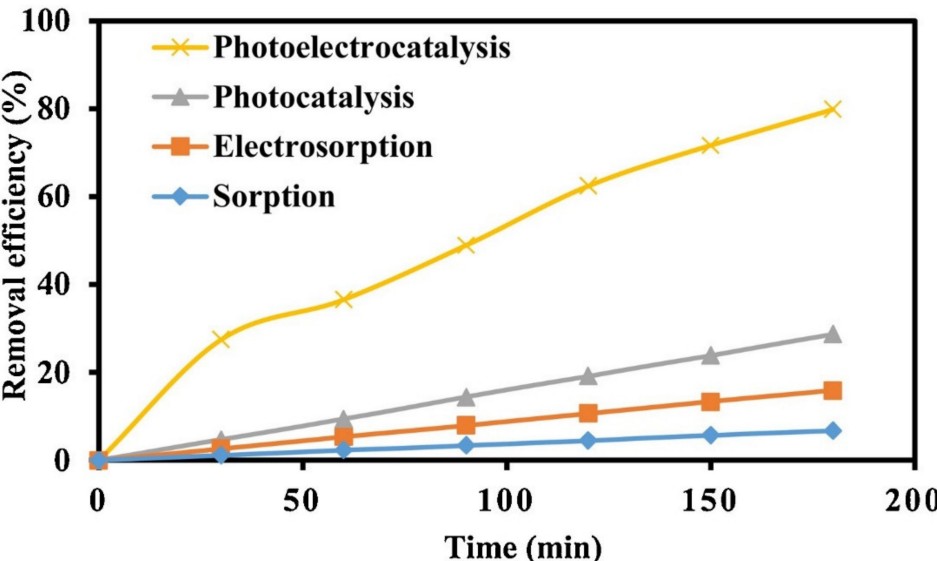

**Figure 23.** The degradation efficiency of cefixime through various treatment processes using the g-C$_3$N$_4$/Ce-ZnO/Ti [103].

## 8. Future Development

Future development on the electrical assisted photocatalytic techniques (i.e., photo-electrocatalytic processes) as advanced oxidation technologies (AOTs) with and without the aids of suitable oxidants is highly required in the near future. Such innovative techniques are mainly applied to effectively remove organic and inorganic pollutants from the wastewaters and waste gases. Any newly developed electrical assisted photocatalytic processes should achieve the following minimum requirements while compared to the traditional processes,

(1) the accomplishment of high removal/oxidation efficiencies of persistent and hazardous pollutants including organic and inorganic compounds,
(2) the preparation of low-cost and long-life photo(electro)catalysts coupled with assisted electrical potential,
(3) the improvement of photo(electro)catalysts' physicochemical properties to achieve high effective resistence of sulfur poisoning,
(4) the achievement of low-voltage requirement and low-energy consumption for the electrical assisted photocatalytic processes, and
(5) the promotion of high rate of redox reactions occurred on the surface of the photo (electro)catalysts.

Furthermore, developing innovative advanced oxidation techniques (AOTs) coupling with photo(electro)catalytic reactions with the aids of chemical oxidants such as ozone are also highly required for the AOTs development. As a matter of fact, successful development of newly AOTs relies on the collaboration of analytical chemists, electrochemists, and engineers to ensure potential application and effective exploration of the aforementioned measures [104]. Nevertheless, many efforts are highly mandatory in the future. To achieve this goal, the real effluents should be tested in practice, the design and construction of optimal photo(electro)catalytic system should be employed by using stable, cheap, and longer service-life electrodes in the industrial scale. Such techno-economic investigations should be performed to improve their viability with respect to traditional techniques.

## 9. Summary

This article overviews the fundamental theories and reaction mechanisms of photocatalytic technologies with and without the assistance of applied electrical potential field for degrading multi-phase organic and inorganic pollutants in the wastewaters and waste gases. Over the past two decades, electrical potential assisted advance oxidation technolo-

gies (E-AOTs) are thought as promising innovative techniques for next generation, because the high treatability of complicated wastewaters and waste gases can be achieved with operative easiness and cost-effectiveness. The potential application of PEC techniques as the emerging E-AOTs to remediate toxic and/or bio-refractory organic and inorganic contaminants in the wastewaters and waste gases has been well-proven and future applicability. The overall oxidation efficiencies of innovative E-AOTs can be easily controlled through automation, and the E-AOTs processes can be compacted with less-space occupation and cost-effective applicability [104].

Based on the up-to-date available developing AOTs, more fundamental studies are urged and required to develop mature PEC technologies with the perspectives to treating real wastewaters and waste gases in the industrial scale in the near future. It is highly required to continuously search for newly developed innovative photocatalysts with the enhanced photo-activity under solar radiation or the illumination of visible light to accelerate the practical photo-activity and removal efficiencies, and enhance the PEC performance on the effective removal of organic and inorganic pollutants in the wastewater and waste gases [1]. These photo(electro)catalysts should be coupled to the suitable pilot-scale reactors to demonstrate the viability of PEC degradation over real wastewaters and waste gases to further scale-up the pilot-scale to the industrial scale. The efficient design and construction of innovative PEC reactors become a future challenge to the environmental scientists, specialists, and engineers, because these newly developed PEC reactors should embody the photocatalytic requirements in the photo(electro)chemical reactors.

**Author Contributions:** C.-S.Y., I.-R.I. and C.-H.H. planned and guided this review article; C.-S.Y., I.-R.I., J.-R.Z., Z.-B.L. and C.-H.S. performed the study; The manuscript was primarily written by C.-S.Y. with the contributions by other co-authors. All authors have read and agreed to the published version of the manuscript.

**Funding:** This study was performed under the auspices of Ministry of Science and Technology (MOST) of ROC (Taiwan) with the research project of MOST- 108-2221-E-110-048-MY3.

**Data Availability Statement:** This review article reported data mainly obtained from previously published literature.

**Acknowledgments:** The authors gratefully acknowledge the financial support from MOST of ROC (Taiwan) and the technological and administrative assistance from Air Pollution Control Laboratory (APCL) in the Institute of Environmental Engineering at National Sun Yat-Sen University. Special thank goes to Associate Professor Chuen-Yn Fan from Department of Applied Foreign Languages at Chia Nan University of Pharmacy and Science for her assistance in editing the revised MS.

**Conflicts of Interest:** The authors declare no conflict of interest.

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
