# Peer review of "A Review of Electrical Assisted Photocatalytic Technologies for the Treatment of Multi-Phase Pollutants"

_catalysts, doi:10.3390/catal11111332_

Round 1
Reviewer 1 Report
This manuscript concerns a comprehensive review article under the title "Electrical Assisted Photocatalytic Technologies for the Treatment of Multi-Phase Pollutants". It is a commendable attempt to describe a brief history of photocatalytic technologies, including fundamental theories and reaction processes, to assist the electrical potential field in the degradation of multi-phase organic and inorganic contaminants in wastewaters and waste gases. The most current advancements and improvements in photocatalytic technologies paired with an external electric field are discussed, including the characterization of nano-sized photo(electro)catalysts and the degradation of multi-phase pollutants. Furthermore, the development of electrically assisted photocatalytic technologies for future applications and future directions in treating organic and inorganic materials in wastewater and waste gases is addressed. This manuscript should be accepted after minor revision as the concept and overall writing are beneficial for the scientific community and industry.
Herein, I am summarizing my concerns.
- Although it contradicts the title, the author writes the word "with/without the assistance of electrical field" (See line, 12).
- Some figures are with low resolution and suggested using a better image, especially figures 4 and 14.
- Most of the time, the author uses lengthy sentences which are hard to understand. (See Line 70-74). Therefore, it is suggested to re-write, where necessary, to support the scientific readership and make it easy to understand.
- The author should consider plagiarism and should reduce it to an acceptable range. Currently, the similarity report shows 18%, and some parts are just copied and pasted.
- If possible, it is suggested to prepare a graphical abstract to understand the review's general idea and attract more audience.
- The author only mentioned the reference number for figures. However, it is recommended to endorse the published work as an example "Reprinted from [80] with permission from Elsevier for copyrights."
- Throughout the manuscript, several typo errors or mistakes are noticed. For example line 198, the word TiO2@rGO should be written as TiO2@rGO. It is highly recommended to revise the manuscript carefully and remove all those minor mistakes to improve the quality.
Author Response
Response to Reviewer 1 Comments
This manuscript concerns a comprehensive review article under the title "Electrical Assisted Photocatalytic Technologies for the Treatment of Multi-Phase Pollutants". It is a commendable attempt to describe a brief history of photocatalytic technologies, including fundamental theories and reaction processes, to assist the electrical potential field in the degradation of multi-phase organic and inorganic contaminants in wastewaters and waste gases. The most current advancements and improvements in photocatalytic technologies paired with an external electric field are discussed, including the characterization of nano-sized photo(electro)catalysts and the degradation of multi-phase pollutants. Furthermore, the development of electrically assisted photocatalytic technologies for future applications and future directions in treating organic and inorganic materials in wastewater and waste gases is addressed. This manuscript should be accepted after minor revision as the concept and overall writing are beneficial for the scientific community and industry.
Herein, I am summarizing my concerns.
Point 1: Although it contradicts the title, the author writes the word "with/without the assistance of electrical field" (See line 12).
Response 1: Thanks for the comment. Although the manuscript focused mainly on reviewing the fundamental theories and reaction mechanisms of photocatalytic technologies with the assistance of electrical field for degrading multi-phase pollutants, some previous literature regarding the photocatalytic technologies without the assistance of electrical field was also reviewed to compare with those with the assistance of electrical field. However, in order to avoid the confusion from the readers, “with/without” has been replaced by “with” per request.
Point 2: Some figures are with low resolution and suggested using a better image, especially Figures 4 and 14.
Response 2: Thanks for the comments. The resolution of Figures 4 and 14 have been improved per request.
Point 3: Most of the time, the author uses lengthy sentences which are hard to understand. (See Line 70-74). Therefore, it is suggested to re-write, where necessary, to support the scientific readership and make it easy to understand.
Response 3: Thanks for the comment. The lengthy sentences in the original Lines 70-74 have been rephrased and made it more concise to improve the scientific readership per request.
Point 4: The authors should consider plagiarism and should reduce it to an acceptable range. Currently, the similarity report shows 18%, and some parts are just copied and pasted.
Response 4: Thanks for the comment. We have rephrased the sentences and revised the paragraphs throughout the entire manuscript accordingly per request, which has reduced the similarity down to 5% or lower for each literature source.
Point 5: If possible, it is suggested to prepare a graphical abstract to understand the review's general idea and attract more audience.
Response 5: Thanks for the comment. A graphical abstract regarding the principal mechanisms of photo(electro)catalytic techniques has been provided per request.
Point 6: The authors only mentioned the reference number for figures. However, it is recommended to endorse the published work as an example "Reprinted from [80] with permission from Elsevier for copyrights."
Response 6: Thanks for the comment. Traditionally, figures are required to be endorsed for the published books, and mostly only reference numbers of literatures are mentioned in the figures for the reviewing papers.
Point 7: Throughout the manuscript, several typo errors or mistakes are noticed. For example Line 198, the word TiO2@rGO should be written as TiO2@rGO. It is highly recommended to revise the manuscript carefully and remove all those minor mistakes to improve the quality.
Response 7: Thanks for the comment. The word “TiO2@rGO” in the manuscript has been replaced by “TiO2@rGO” per request. Also, the typo errors/mistakes have been revised throughout the entire manuscript per request.

Reviewer 2 Report
The manuscript with the ID catalysts-1424687, entitled ’’A Review of Electrical Assisted Photocatalytic Technologies for the Treatment of Multi-Phase Pollutants” is presenting a nice piece critical review in the filed of electrical assisted photocatalysis. Basically, the principles and advantages of photo(electro)catalysis are well highlighted. Nevertheless, the following minor changes are needed before this manuscript could be accepted:
-Page 2, lines 57-95: In the Introduction, the efficiency of photo(electro)catalysis in removing organic pollutants is insufficiently argued, especially comparing with other advanced AOPs processes like sono-photo-catalysis.
-Page 12, lines 367-368: Please give some examples (with cited references) for the removal of organic pollutants using TiO2 as photocatalyst.
-Page 20, lines 667-684: Please insert a new figure (or at least please mention the degradation products in words) for the photoelectrocatalytic oxidation of phenol.
Author Response
Response to Reviewer 2 Comments
The manuscript with the ID catalysts-1424687, entitled “A Review of Electrical Assisted Photocatalytic Technologies for the Treatment of Multi-Phase Pollutants” is presenting a nice piece critical review in the field of electrical assisted photocatalysis. Basically, the principles and advantages of photo(electro)catalysis are well highlighted. Nevertheless, the following minor changes are needed before this manuscript could be accepted:
Point 1: Page 2, lines 57-95: In the Introduction, the efficiency of photo(electro)catalysis in removing organic pollutants is insufficiently argued, especially comparing with other advanced AOPs processes like sono-photo-catalysis.
Response 1: Thanks for the comments. The degradation efficiencies of organic pollutants by photo(electro)catalysis have been compared with other AOPs in the manuscript. Particularly, the principle of sono-photo-catalysis has been reported in Pages 2-3, lines 97-107 per request.
Point 2: Page 12, lines 367-368: Please give some examples (with cited references) for the removal of organic pollutants using TiO2 as photocatalyst.
Response 2: Thanks for the comments. The example have already mentioned in Page 26-27, lines 874-921 per request.
Point 3: Page 20, lines 667-684: Please insert a new figure (or at least please mention the degradation products in words) for the photoelectrocatalytic oxidation of phenol.
Response 3: Thanks for the comment. A new figure (Figure 13) has been inserted in the sentence which was added to the manuscript per request as shown below,
“Previous study reported that phenol can be converted to aromatic, then decomposed to aliphatic, and finally oxidized to CO2. The oxidation processes are illustrated in Figure 13 [91]”.
